



# Regional modeling of internal-tide dynamics around New Caledonia. Part 2: Tidal incoherence and implications for sea surface height observability

Arne Bendinger[1,a], Sophie Cravatte[1,3], Lionel Gourdeau[1], Clément Vic[2], and Florent Lyard[1]

[1]Université de Toulouse, LEGOS (CNES/CNRS/IRD/UT3), Toulouse, France
[2]Laboratoire d'Océanographie Physique et Spatiale, Univ. Brest, CNRS, Ifremer, IRD, IUEM, Brest, France
[3]IRD, Centre IRD de Nouméa, New Caledonia
[a]now at: Laboratoire d'Océanographie Physique et Spatiale, Univ. Brest, CNRS, Ifremer, IRD, IUEM, Brest, France

**Correspondence:** Arne Bendinger (arne.bendinger@univ-brest.fr)

**Abstract.** New Caledonia, in the southwestern tropical Pacific, has recently been identified as a hot spot for energetic semidiurnal internal tides. In a companion paper, the life cycle of coherent internal tides, characterized by fixed amplitude and phase, was investigated in the regions through harmonic analysis of a year-long, hourly time series from numerical simulation output. In this study, we investigate temporal variability of the internal tide by decomposing the semidiurnal signals into coherent and incoherent components. We show that the departure from coherence, in both generation and propagation of internal tides, is linked to the presence of mesoscale eddies and attempt to identify the underlying mechanisms. Our results suggest that temporal variability of the barotropic-to-baroclinic energy conversion is largely governed by the coherent component (>90%) through the astronomically forced fortnightly modulated spring-neap cycle, which induces biweekly conversion variations by a factor between 3 and 7. The incoherent component - negligible in the annual mean - can explain on monthly to intraseasonal scales a notable fraction of variability while reducing or enhancing semidiurnal conversion by up to 20 %. We find that it is dominated by local effects such as the work of the barotropic tide on baroclinic bottom pressure amplitude variations, and linked to mesoscale-eddy-induced stratification changes. Away from the generation sites, tidal incoherence becomes increasingly important, manifesting as refraction and altered orientation of tidal beams due to interactions with mesoscale currents and varying group and phase speeds. The incoherent sea surface height signature, with a root-mean-square amplitude of 1–2 cm, is widespread across the domain. In tidal beam propagation direction, this incoherent component introduces limitations in the spectral observability of mesoscale to submesoscale dynamics below 80 km wavelength. Along altimetry tracks that are not aligned with the predominant propagation direction, the incoherent tide tends to dominate over the coherent tide due to its more isotropic nature. Consequently, transition scales estimating the wavelength at which balanced motion becomes dominant over unbalanced motion should be interpreted with caution in regions with pronounced internal-tide activity and well-defined propagation directions.





## 1 Introduction

Internal tides, i.e., internal waves at tidal frequency, are ubiquitous in the global ocean, where major bathymetric obstacles cause the energy transfer from the barotropic tide to baroclinic waves. These internal waves may either dissipate locally or propagate towards the open ocean: they are argued to play an essential role in our understanding of open-ocean mixing and the global oceanic energy budget (Melet et al., 2013; Waterhouse et al., 2014; Melet et al., 2016; de Lavergne et al., 2022). Well-known internal-tide generation sites for low-vertical modes include Luzon Strait (e.g. Alford et al., 2011; Rainville et al., 2013; Kerry et al., 2016; Wang et al., 2023), the Hawaiian Ridge (e.g. Merrifield and Holloway, 2002; Carter et al., 2008; Zilberman et al., 2011), the Indonesian/Solomon Seas (e.g. Nagai and Hibiya, 2015; Tchilibou et al., 2020), and the Amazonian Shelf Break (e.g. Tchilibou et al., 2022). From semianalytical theory (Falahat et al., 2014b; de Lavergne et al., 2019; Vic et al., 2019), satellite altimetry (Ray and Zaron, 2016; Zhao et al., 2016; Zaron, 2019), and global numerical simulations (Müller et al., 2012; Shriver et al., 2012; Arbic, 2022; Buijsman et al., 2017), the southwestern tropical Pacific is also known to be an internal-tide generation hot spot, but has not received particular attention until very recently (Bendinger et al., 2023, 2024).

Based on a tailored regional numerical modeling effort and a full-calendar year, hourly time series simulation output, Bendinger et al. (2023, hereinafter Part 1) characterized the internal-tide life cycle around New Caledonia, an archipelago at the entrance of the Coral Sea and an internal-tide generation hot spot of the semidiurnal M2 tide. They found that a total of 15.27 GW is converted from the barotropic to baroclinic tide, closely associated with the northwest-southeast-extending ridge system, continental slopes, shelf breaks, and seamounts. More than 50 % of the locally generated internal tides were found to dissipate close to the generation sites, suggesting a major contribution to vertical mixing processes. The remaining energy propagates in well-defined low-vertical mode tidal beams north and south of New Caledonia towards the open ocean with depth-integrated baroclinic energy fluxes of up to 30 kW m$^{-1}$. Overall, these tidal energetics are well comparable with other prominent internal-tide generation sites in the Pacific such as Luzon Strait or the Hawaiian Ridge.

One limitation of the previous analysis in Part 1 lies in the applied harmonic analysis on which the above results are based and which is representative of the coherent tide only. The term coherent refers here to tidal stationarity, i.e., fixed amplitude and phase for a given location, in-phase with the astronomical tide forcing, and hence, well predictable in space and time. Further, the incoherent tide, i.e., temporal variability, was not addressed. Tidal incoherence is defined as any departure from tidal coherence, i.e., temporal variations of amplitude and phase within the tidal frequency band, and therefore characterized by unpredictability (Nash et al., 2012; Kerry et al., 2013; Buijsman et al., 2017). It can represent an essential fraction of the total variance and an important contribution to the internal-tide's life cycle (Zilberman et al., 2011; Buijsman et al., 2017; Zaron, 2017). As a consequence, internal-tide dissipation - estimated as the residual between local conversion and energy flux divergence and deduced in Part 1 - may have been overestimated as it contained both true baroclinic energy dissipation and energy transferred to the incoherent tide. In addition, some of the conclusions, such as those on sea surface height (SSH), should be



revisited while considering the importance of tidal incoherence.


Sources of tidal incoherence are numerous and linked to interannual, seasonal, and mesoscale variability of stratification and background currents (Rainville and Pinkel, 2006; Tchilibou et al., 2020; Cai et al., 2024; Kaur et al., 2024). In the near-field, i.e., close to internal-tide generation sites, local stratification changes or impinging remotely generated internal tides can induce barotropic-to-baroclinic conversion variations by altering the baroclinic bottom pressure (Kelly and Nash, 2010; Zilberman

et al., 2011; Kerry et al., 2014; Pickering et al., 2015; Kerry et al., 2016). In the far-field, i.e., away from the generation site and in tidal energy propagation direction, tidal incoherence is commonly induced by spatiotemporal variations of stratification and background currents such as mid-latitude and equatorial jets as well as the mesoscale eddy field (Park and Watts, 2006; Rainville and Pinkel, 2006; Dunphy and Lamb, 2014; Ponte and Klein, 2015; Kelly and Lermusiaux, 2016; Buijsman et al., 2017; Duda et al., 2018; Guo et al., 2023). The mechanisms responsible for the internal-tide's temporal variability and their

governing dynamics are geographically highly variable. As such, the responsible mechanisms cannot be generalized, as the importance of seasonality (among others) is not directly comparable across ocean basins. Further, temporal variations may be highly unpredictable, linked to mesoscale eddy variability. Quantifying these dynamics remains a challenge and demands dedicated regional studies. New Caledonia is a particularly challenging region being both a hot spot of internal-tide generation and mesoscale variability, potentially subject to eddy-internal tide interactions.


Whether tidal incoherence facilitates tidal energy to dissipate is of current research interest. Generally, a loss of tidal coherence is not directly linked to tidal energy dissipation, but rather with non-linear energy transfer to the incoherent tide. Though, it is often associated with energy being transferred from low to higher vertical modes with the latter that tend to dissipate (Dunphy and Lamb, 2014; Bella et al., 2024). Wang and Legg (2023) explicitly showed that baroclinic eddies are capable of trapping

low-mode internal tides and effectively transport energy to higher-vertical modes, resulting in internal-tide dissipation within eddies. However, a robust estimate of the fraction of energy dissipation associated with such processes does not exist and may also vary from region to region due to the different underlying dynamics. In turn, this can have important implications for tidal-mixing parameterizations in climate and ocean general circulation models, which do not resolve tidal processes. Current parameterizations rely heavily on semi-analytical theory, which do not take into account temporal variations of the background

stratification and currents (Vic et al., 2019; de Lavergne et al., 2019, 2020). Further effort is therefore needed to quantify the internal-tide life cycle and energy dissipation.

Internal tides are of particular interest in the context of the Surface Water Ocean Topography (SWOT) satellite altimetry mission launched in December 2022 and providing high-resolution SSH observations down to 15 km wavelength - an order of

magnitude higher resolution than conventional satellite altimetry - for three main reasons (Fu and Ubelmann, 2014; Fu et al., 2024). First, SWOT will contribute to the characterization of the internal-tide SSH signature including those associated with higher vertical modes, which are associated with shorter horizontal scales than low vertical modes (Arbic et al., 2018). Second, combined SSH observations of balanced and unbalanced motions allow us to study non-linear scale interactions down





to submesoscales and eventually help understand the energy transfer toward dissipation scales on global scales (Klein et al.,
2019; Morrow et al., 2019). Third, accurate knowledge of the internal-tide SSH signature is crucial for the study of mesoscale
and submesoscale dynamics. In other words, the extent to which we can exploit SWOT's potential to study fine-scale oceanic
dynamics depends on our ability to disentangle the measured SSH signal associated with balanced and unbalanced motions,
i.e., mesoscale to submesoscale dynamics and internal gravity waves. In this regard, the transition scale serves typically as a
quantity to estimate the length scale below unbalanced motions become dominant over balanced motions (Vergara et al., 2023).
Though, this is problematic in regions where balanced and unbalanced motions feature equal SSH variance at similar wave-
lengths (Callies and Wu, 2019). A correction for the coherent internal tide partially addresses this problem while increasing
SSH observability of mesoscale to submesoscale dynamics (Qiu et al., 2018; Carrere et al., 2021; Arbic, 2022). The empirical
High-Resolution Empirical Tide (HRET) model derived from the over 20-year long altimeter time-series is currently used to
apply such correction (Zaron, 2019). However, this correction does not consider incoherent internal tides that can explain a
large fraction of the total SSH variance (Zaron, 2017; Lahaye et al., 2024). This makes the dynamical interpretation of SWOT
SSH, i.e., the allocation of different dynamics to SSH, a difficult challenge in regions where tidal incoherence is important.

The objective of this companion paper is to (1) identify regions of important tidal incoherence around New Caledonia, (2)
quantify its relative importance across different time scales, and (3) understand the underlying mechanisms leading to tidal in-
coherence. What are the processes, both in the near- and far-field, that drive temporal variability in the barotropic-to-baroclinic
conversion term and the propagation of energy? What insights can be gained regarding the temporal variability of internal-tide
dissipation? Finally, (4) how does the incoherent tide manifest in SSH and what are potential implications for SSH observ-
ability of balanced motions in New Caledonia - a region where disentangling SWOT SSH from different dynamics presents
significant challenges?


The study is organized as follows. In Sect. 2, we give a description of the regional numerical model configuration, previously
introduced in Part 1, as well as the underlying methodology used to infer temporal variability of the internal tide. In Sect. 3,
we begin with an analysis of the semidiurnal tidal diagnostics, decomposed into their coherent and incoherent components.
We then attribute in greater detail how mesoscale eddy variability drives temporal variability in the barotropic-to-baroclinic
conversion term and energy flux. In Sect. 6 we address the importance of the incoherent SSH signature around New Caledonia
and what implications it can have for SSH observability of mesoscale to submesoscale dynamics. We finish with a summary
and perspectives of this work in Sect. 7.





## 2 Data and methods

### 2.1 Numerical simulation

We use a regional model configuration that consists of a host grid (TROPICO12, 1/12°horizontal resolution and 125 vertical levels) that covers the tropical and subtropical Pacific Ocean basin from 142° E-70° W and 46° S-24° N, introduced in Part 1. The oceanic reanalysis GLORYS2V4 prescribes initial conditions for temperature and salinity as well as the forcing with daily currents, temperature, and salinity at the open lateral boundaries. ERA5 produced by the European Centre for Medium-Range Weather Forecasts (ECMWF, Hersbach et al., 2020) provides atmospheric forcing at hourly temporal resolution and a spatial

resolution of 1/4°to compute surface fluxes using bulk formulae and the model prognostic sea surface temperature. The model is forced by the tidal potential of the five major diurnal (K1, O1) and semidiurnal (M2, S2, N2) tidal constituents. At the open lateral boundaries it is forced by barotropic SSH and barotropic currents of the same five tidal constituents taken from the global tide atlas FES2014 (Finite Element Solution 2014, Lyard et al., 2021).

We build on hourly numerical simulation outputs from a higher-resolution horizontal grid refinement within the host grid in the southwestern tropical Pacific Ocean encompassing New Caledonia (CALEDO60). This nesting grid features 1/60°horizontal resolution or ∼1.7 km grid spacing initialized by an Adaptive Grid Refinement in Fortran (AGRIF, Debreu et al., 2008). Specifically, we make use of the three-dimensional velocity, temperature, salinity, and pressure fields as well as SSH. Coherent tidal harmonics are taken from Part 1, based on harmonic analysis and vertical mode decomposition. We refer to Sect. 2.2

in Part 1 for more details. Both, full-model variables and tidal harmonics are constrained to a full-calendar year (model year 2014). Constraining the analysis to a full-calendar year relies upon a compromise between high computational expenses and a time series long enough for a robust extraction of the coherent tide.

The model's eligibility of realistically simulating ocean dynamics that range from the large-scale circulation down to high-frequency motion using climatology, satellite altimetry, and in-situ measurements was addressed in Part 1. Regarding the

dominant semidiurnal M2 internal tide, confidence in the model's SSH signature is explicitly given by comparison with HRET revealing reasonable amplitude and large-scale patterns (see their Fig. 13a-d). Furthermore, glider observations revealed the model's accurate representation of the spatiotemporal variability of the semidiurnal energy flux south of New Caledonia, i.e., the location, magnitude, and vertical structure/extent of the westward internal tide energy propagation characterized by narrow tidal beams (Bendinger et al., 2024).

### 2.2 Tidal analysis and diagnostics

The internal-tide life cycle, i.e., internal-tide generation, propagation, and dissipation, is typically studied using the following depth-integrated baroclinic energy equation:

$$T + \boldsymbol{\nabla}_{\mathrm{h}} \cdot \mathbf{F}_{\mathrm{bc}} + A = C - D_{\mathrm{bc}}, \tag{1}$$

... 



where $T = \partial/\partial t \left( \int_{-H}^{0} (KE_{\mathrm{bc}} + APE) dz \right)$ is the tendency term of the total (baroclinic kinetic energy $KE_{\mathrm{bc}}$ and available

potential energy $APE$), $\boldsymbol{\nabla}_{\mathrm{h}} \cdot \mathbf{F}_{\mathrm{bc}}$ is the baroclinic energy flux divergence with $\boldsymbol{\nabla}_{\mathrm{h}}$ the horizontal gradient operator and $\mathbf{F}_{\mathrm{bc}} = (F_x, F_y)$ the energy flux vector, $A = \int_{-H}^{0} \boldsymbol{\nabla}_{\mathrm{h}} \cdot (\mathbf{u}(KE_{\mathrm{bc}} + APE + \rho_0 \mathbf{u}_{\mathrm{bt}} \cdot \mathbf{u}_{\mathrm{bc}})) dz$ the advection term, where $\mathbf{u} = \mathbf{u}_{\mathrm{bt}} + \mathbf{u}_{\mathrm{bc}}$

is the horizontal velocity vector decomposed into its barotropic ($_{\mathrm{bt}}$) and baroclinic($_{\mathrm{bc}}$) parts, and $\rho_0$ the reference density. $C$ is the barotropic-to-baroclinic conversion term as further defined below, and $D_{\mathrm{bc}}$ baroclinic energy dissipation. Following Simmons et al. (2004), Carter et al. (2008), Buijsman et al. (2014, 2017), and Lahaye et al. (2020), we investigated in Part 1 the depth-integrated baroclinic energy budget by neglecting the tendency and advection terms, i.e.,:

$$\boldsymbol{\nabla}_{\mathrm{h}} \cdot \mathbf{F}_{\mathrm{bc}} - C + D_{\mathrm{bc}} = 0. \tag{2}$$

This simplification has been made since tendency and advection terms are negligibly small in the above studies when averaged over a set of tidal periods and when compared to $\boldsymbol{\nabla}_{\mathrm{h}} \cdot \mathbf{F}_{\mathrm{bc}}$ (Kang and Fringer, 2012). Nonetheless, and strictly speaking, $D_{\mathrm{bc}}$ should here be regarded as a residual of energy flux divergence and conversion. In Part 1, we estimated the coherent M2 internal tide energy budget based on a harmonic analysis referenced to a full-model calendar year. Here, in Part 2 our major objective is to deduce time variability to Equation 2 for the semidiurnal frequency band ($^{\mathrm{D2}}$, 10-14 h) through a bandpass-filtering technique following Nash et al. (2012), Pickering et al. (2015), and Buijsman et al. (2017). Specifically, time variability is inferred by decomposing the bandpass-filtered contribution in Equation 2 into the coherent ($^{\mathrm{coh}}$) and incoherent ($^{\mathrm{inc}}$) parts, i.e., $\mathbf{u}$ and the pressure perturbation $p$:

$$\mathbf{u}^{\mathrm{D2}} = \mathbf{u}^{\mathrm{coh}} + \mathbf{u}^{\mathrm{inc}}, \quad p^{\mathrm{D2}} = p^{\mathrm{coh}} + p^{\mathrm{inc}}. \tag{3}$$

$\mathbf{u}^{\mathrm{D2}}$ and $p^{\mathrm{D2}}$ can be further decomposed into their barotropic and baroclinic parts. The former is estimated by the depth-average and the latter by the departure from the barotropic depth-average:

$$\mathbf{u}_{\mathrm{bc}}^{\mathrm{D2}} = \mathbf{u}^{\mathrm{D2}} - \underbrace{\frac{1}{H} \int\limits_{-H}^{0} \mathbf{u}^{\mathrm{D2}} dz}_{\mathbf{u}_{\mathrm{bt}}^{\mathrm{D2}}}, \quad p_{\mathrm{bc}}^{\mathrm{D2}} = p^{\mathrm{D2}} - \underbrace{\frac{1}{H} \int\limits_{-H}^{0} p^{\mathrm{D2}} dz}_{p_{\mathrm{bt}}^{\mathrm{D2}}}, \tag{4}$$

where $H$ is the water column depth. The semidiurnal conversion term $C^{\mathrm{D2}}$ can then be written as:

$$C^{\mathrm{D2}} = \frac{1}{T} \int\limits_{T} -\boldsymbol{\nabla}_{\mathrm{h}} H \cdot \mathbf{u}_{\mathrm{bt}}^{\mathrm{D2}} p_{\mathrm{bc}}^{\mathrm{D2}}(-H) dt \tag{5}$$

$$= \frac{1}{T} \int\limits_{T} (\underbrace{-\boldsymbol{\nabla}_{\mathrm{h}} H \cdot \mathbf{u}_{\mathrm{bt}}^{\mathrm{coh}} p_{\mathrm{bc}}^{\mathrm{coh}}(-H)}_{C^{\mathrm{coh}}} \underbrace{-\boldsymbol{\nabla}_{\mathrm{h}} H \cdot \mathbf{u}_{\mathrm{bt}}^{\mathrm{inc}} p_{\mathrm{bc}}^{\mathrm{inc}}(-H)}_{C^{\mathrm{inc*}}} \underbrace{-\boldsymbol{\nabla}_{\mathrm{h}} H \cdot \mathbf{u}_{\mathrm{bt}}^{\mathrm{coh}} p_{\mathrm{bc}}^{\mathrm{inc}}(-H) - \boldsymbol{\nabla}_{\mathrm{h}} H \cdot \mathbf{u}_{\mathrm{bt}}^{\mathrm{inc}} p_{\mathrm{bc}}^{\mathrm{coh}}(-H)}_{C^{\mathrm{cross}}}) dt, \tag{6}$$

$$\underbrace{\phantom{-\boldsymbol{\nabla}_{\mathrm{h}} H \cdot \mathbf{u}_{\mathrm{bt}}^{\mathrm{inc}} p_{\mathrm{bc}}^{\mathrm{inc}}(-H) -\boldsymbol{\nabla}_{\mathrm{h}} H \cdot \mathbf{u}_{\mathrm{bt}}^{\mathrm{coh}} p_{\mathrm{bc}}^{\mathrm{inc}}(-H) - \boldsymbol{\nabla}_{\mathrm{h}} H \cdot \mathbf{u}_{\mathrm{bt}}^{\mathrm{inc}} p_{\mathrm{bc}}^{\mathrm{coh}}(-H)}}_{C^{\mathrm{inc}}}$$



decomposed into the coherent ($C^{\mathrm{coh}}$) and incoherent conversion ($C^{\mathrm{inc}}$), averaged over a given time period $T$. Note that we consider the cross terms as incoherent (i.e., $C^{\mathrm{inc}} = C^{\mathrm{inc}^*} + C^{\mathrm{cross}}$) since there is no preferred phasing between coherent and incoherent motions (Nash et al., 2012). Further, we define $C^{\mathrm{cross1}} = -\boldsymbol{\nabla}_{\mathrm{h}}H\cdot\mathbf{u}^{\mathrm{coh}}_{\mathrm{bt}}p^{\mathrm{inc}}_{\mathrm{bc}}(-H)$ and $C^{\mathrm{cross2}} = -\boldsymbol{\nabla}_{\mathrm{h}}H\cdot\mathbf{u}^{\mathrm{inc}}_{\mathrm{bt}}p^{\mathrm{coh}}_{\mathrm{bc}}(-H)$.

In this study, we put a particular focus on the importance of each of these terms to better understand temporal variations of the conversion term and the underlying governing processes. $C^{\mathrm{inc}}$ is typically considered to be negligible when averaged over long time periods (such as the annual mean) since the incoherent component of the barotropic tide forcing ($\mathbf{u}^{\mathrm{inc}}_{\mathrm{bt}}$) is assumed to be small on the considered time scales. Further, the unpredictable nature and random phase of incoherent processes or the interference of coherent and incoherent motions cancel out in the long-term (Nash et al., 2012; Buijsman et al., 2017). How-

ever, we will show in the analysis below that these terms are non-negligible on shorter time scales and, in fact, do explain an important fraction of the variability.

Similarly to $C^{\mathrm{D2}}$, the semidiurnal depth-integrated energy flux $\mathbf{F}^{\mathrm{D2}}_{\mathrm{bc}}$ can be written as:

$$\mathbf{F}^{\mathrm{D2}}_{\mathrm{bc}} = \frac{1}{T}\int\limits_{T}\int\limits_{-H}^{0}\mathbf{u}^{\mathrm{D2}}_{\mathrm{bc}}p^{\mathrm{D2}}_{\mathrm{bc}}\,dz\,dt \tag{7}$$

$$= \frac{1}{T}\int\limits_{T}\left(\underbrace{\int\limits_{-H}^{0}\mathbf{u}^{\mathrm{coh}}_{\mathrm{bc}}p^{\mathrm{coh}}_{\mathrm{bc}}\,dz}_{F^{\mathrm{coh}}_{\mathrm{bc}}} + \underbrace{\int\limits_{-H}^{0}(\mathbf{u}^{\mathrm{inc}}_{\mathrm{bc}}p^{\mathrm{inc}}_{\mathrm{bc}} + \mathbf{u}^{\mathrm{coh}}_{\mathrm{bc}}p^{\mathrm{inc}}_{\mathrm{bc}} + \mathbf{u}^{\mathrm{inc}}_{\mathrm{bc}}p^{\mathrm{coh}}_{\mathrm{bc}})\,dz}_{F^{\mathrm{inc}}_{\mathrm{bc}}}\right)dt. \tag{8}$$

The coherent terms $\mathbf{u}^{\mathrm{coh}}_{\mathrm{bt}}$, $\mathbf{u}^{\mathrm{coh}}_{\mathrm{bc}}$, and $p^{\mathrm{coh}}_{\mathrm{bc}}$ are taken from the tidal analysis and the underlying harmonic analysis and vertical mode decomposition carried out in Part 1. Here, they are representative of the semidiurnal frequency band, i.e., the sum of M2, S2, and N2. The barotropic part is estimated by mode 0 and the baroclinic part by the sum of modes 1-9. The incoherent terms

are computed as the difference between the above semidiurnal bandpassed and coherent time series, i.e.,

$$\mathbf{u}^{\mathrm{inc}}_{\mathrm{bt}} = \mathbf{u}^{\mathrm{D2}}_{\mathrm{bt}} - \mathbf{u}^{\mathrm{coh}}_{\mathrm{bt}}, \quad \mathbf{u}^{\mathrm{inc}}_{\mathrm{bc}} = \mathbf{u}^{\mathrm{D2}}_{\mathrm{bc}} - \mathbf{u}^{\mathrm{coh}}_{\mathrm{bc}}, \quad p^{\mathrm{inc}}_{\mathrm{bc}} = p^{\mathrm{D2}}_{\mathrm{bc}} - p^{\mathrm{coh}}_{\mathrm{bc}}. \tag{9}$$

Dissipation is estimated through the residual between energy flux divergence and conversion as discussed above. As such, semidiurnal energy dissipation is defined by:

$$D^{\mathrm{D2}}_{\mathrm{bc}} = \frac{1}{T}\int\limits_{T}(-\boldsymbol{\nabla}_{\mathrm{h}}\cdot\mathbf{F}^{\mathrm{D2}}_{\mathrm{bc}} + C^{\mathrm{D2}})\,dt \tag{10}$$

$$= \frac{1}{T}\int\limits_{T}(\underbrace{-\boldsymbol{\nabla}_{\mathrm{h}}\cdot\mathbf{F}^{\mathrm{coh}}_{\mathrm{bc}} + C^{\mathrm{coh}}}_{D^{\mathrm{coh}}_{\mathrm{bc}}} + \frac{1}{T}\int\limits_{T}(\underbrace{-\boldsymbol{\nabla}_{\mathrm{h}}\cdot\mathbf{F}^{\mathrm{inc}}_{\mathrm{bc}} + C^{\mathrm{inc}}}_{D^{\mathrm{inc}}_{\mathrm{bc}}})\,dt, \tag{11}$$





assuming that the contributions from the tendency and advection terms are negligibly small. A worth noting conclusion concerning the coherent internal-tide analysis in Part 1 was that $D_{\mathrm{bc}}^{\mathrm{coh}}$ may not be associated entirely with true energy dissipation. In fact, $D_{\mathrm{bc}}^{\mathrm{coh}}$ represents both energy dissipation and energy transfer to the incoherent tide. It also includes the energy transfer to higher harmonics (higher frequency waves) which, however, expresses presumably in numerical dissipation due to spatial resolution constraints (Peacock and Tabaei, 2005; Zeng et al., 2021). While $D_{\mathrm{bc}}^{\mathrm{D2}}$ accounts for actual energy dissipation, $D_{\mathrm{bc}}^{\mathrm{inc}}$ can be understood as the fraction by which $D_{\mathrm{bc}}^{\mathrm{coh}}$ is mistakenly associated with true energy dissipation. In other words, it represents the error by which energy dissipation in $D_{\mathrm{bc}}^{\mathrm{coh}}$ is overestimated.

## 2.3 Ray tracing

A ray tracing method following Rainville and Pinkel (2006) is used to interpret the departure from tidal coherence in the far-field, associated with the refraction of the tidal beam due to mesoscale background currents, i.e., mesoscale eddies. Specifically, the theoretical horizontal propagation of internal gravity modes is investigated considering spatially varying topography, climatological stratification, planetary vorticity, and depth-independent currents. Following this approach, we assume that departure from tidal coherence is primarily due to varying background currents. The choice of depth-independent currents relies on the general assumption that mesoscale eddies are well represented by a barotropic and a mode-1 baroclinic structure with limited vertical shear (Smith and Vallis, 2001).

Bathymetry is taken from ETOPO2v2 (Smith and Sandwell, 1997). Internal gravity wave speeds are predefined and solved by the Sturm-Liouville problem for stratification from the World Ocean Atlas (Locarnini et al., 2018; Zweng et al., 2019). We model semidiurnal ray paths for modes 1-2, initialized at 167.75° E, 18.4° S and 167.65° E, 23.35° S within the two major internal-tide generation hot spots as identified in Part 1, i.e., the North (1) and South (2) domains (see Fig. 1, and for a given propagation angle: northeastward (45°) and southwestward (210°), respectively. In an iterative procedure, the ray tracing considers for each step size (1 km) bathymetry, climatological buoyancy and planetary vorticity effects, and the background currents. Through the dispersion relation from the Helmholtz equation for internal wave modes assuming a local wave expression, the ray's group and phase velocity are obtained, which are then used to update the ray's position and direction (angle of propagation). To mimic the effects of background currents on the ray's path, a no-currents scenario is also given. The ray tracing is applied on the depth-averaged currents as derived from the daily-mean three-dimensional velocity field from the regional model output (CALEDO60). The eligibility of the simplified ray tracing's methodology to infer tidal beam refraction was demonstrated in Bendinger et al. (2024). Specifically, the assumption of considering only depth-independent currents is validated a posteriori, given the relevance of the qualitative picture of ray trajectories that are obtained.

## 3 Annual mean of semidiurnal, coherent, and incoherent tidal diagnostics

The depth-integrated semidiurnal barotropic-to-baroclinic conversion, energy flux, and dissipation (residual) were computed for a full-model calendar year and in the full-regional domain, decomposed into the coherent and incoherent components



(see Sect. 2.2). The annual mean is shown in Fig. 1. In Part 1, we identified four hot-spot regions of internal-tide generation: North (1), South (2), Norfolk Ridge (3), and Loyalty Ridge (4) (see Fig. 1). Integrated over the subdomains, the barotropic-to-

baroclinic conversion is almost entirely dominated by the coherent component (Fig. 1a-b, Table 1). The incoherent component is negligible (Fig. 1c). However, we will show in Sect. 4 that on shorter time scales the conversion term is subject to temporal variations not linked to the astronomical tide forcing.

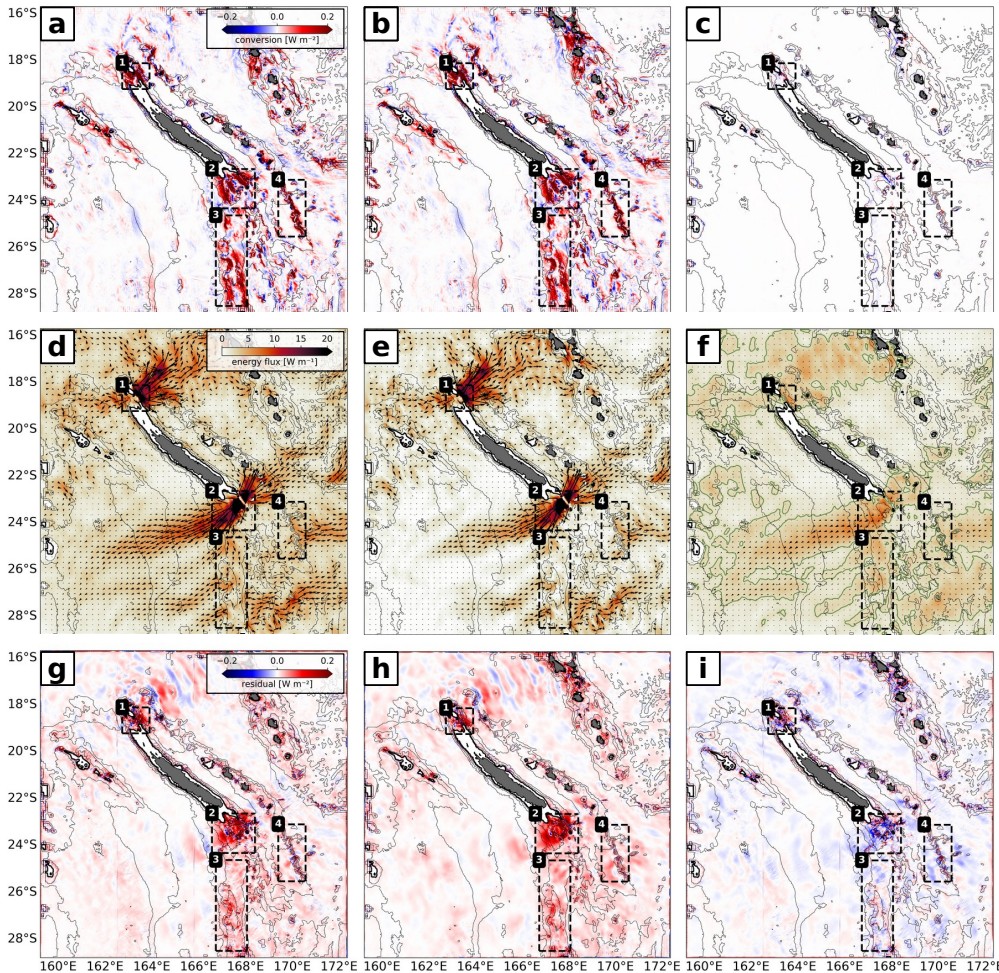

**Figure 1.** Annual mean, depth-integrated, semidiurnal (a) barotropic-to-baroclinic energy conversion, decomposed into the (b) coherent, and (c) incoherent components. (d-f) and (g-i) same as (a-c) but for the depth-integrated, semidiurnal energy flux and dissipation (residual), respectively. For the incoherent energy flux (f), the 2 kW m$^{-1}$ contour is also shown. The thin black lines represent the 1000, 2000, and 3000 m depth contours. The thick black line is the 100 m depth contour representative of the New Caledonian lagoon. The numbered black boxes represent the hot spots of internal-tide generation (1: North, 2: South, 3: Norfolk Ridge, 4: Loyalty Ridge).





**Table 1.** Annual mean and standard deviation of the regional semidiurnal barotropic-to-baroclinic conversion $C^{\mathrm{D2}}$, baroclinic energy flux divergence $\nabla F_{\mathrm{bc}}^{\mathrm{D2}}$, and baroclinic dissipation $D_{\mathrm{bc}}^{\mathrm{D2}}$ integrated over the North (1), South (2), Norfolk Ridge (3), Loyalty Ridge (4) domains, and decomposed into their coherent and incoherent parts.

|  |  | North | South | Norfolk Ridge | Loyalty Ridge |
|---|---|---|---|---|---|
| $C^{\mathrm{D2}}$ | semidiurnal | $2.79 \pm 1.56$ | $4.90 \pm 2.58$ | $2.45 \pm 1.13$ | $1.23 \pm 0.60$ |
|  | coherent | $2.81 \pm 1.57$ | $4.97 \pm 2.62$ | $2.47 \pm 1.13$ | $1.19 \pm 0.56$ |
|  | incoherent | $-0.03 \pm 0.32$ | $-0.07 \pm 0.69$ | $-0.02 \pm 0.42$ | $0.04 \pm 0.20$ |
| $\boldsymbol{\nabla}_{\mathrm{h}} \cdot \mathbf{F}_{\mathrm{bc}}^{\mathrm{D2}}$ | semidiurnal | $1.72 \pm 0.95$ | $2.06 \pm 1.11$ | $1.47 \pm 0.72$ | $0.89 \pm 0.46$ |
|  | coherent | $1.78 \pm 0.97$ | $1.80 \pm 0.88$ | $1.41 \pm 0.61$ | $0.75 \pm 0.35$ |
|  | incoherent | $-0.06 \pm 0.19$ | $0.26 \pm 0.46$ | $0.06 \pm 0.28$ | $0.14 \pm 0.26$ |
| $D_{\mathrm{bc}}^{\mathrm{D2}}$ | semidiurnal | $1.06 \pm 0.67$ | $2.83 \pm 1.57$ | $0.97 \pm 0.48$ | $0.35 \pm 0.28$ |
|  | coherent | $1.03 \pm 0.62$ | $3.17 \pm 1.76$ | $1.06 \pm 0.55$ | $0.45 \pm 0.21$ |
|  | incoherent | $0.03 \pm 0.31$ | $-0.33 \pm 0.54$ | $-0.09 \pm 0.31$ | $-0.10 \pm 0.21$ |

All units are given in [GW].

As stated in Part 1, the depth-integrated energy flux is characterized in the annual mean by two tidal beams that emerge and diverge from the North (1) and South (2) domains (Fig. 1d). While the coherent component is dominant, the incoherent component does explain an important fraction (Fig. 1e-f). This is particularly the case for the South (2) and Loyalty Ridge (4) domains, where the area-integrated incoherent energy flux divergence accounts for roughly 13 % and 16 % of the semidiurnal energy flux divergence, respectively (Table 1). In the North (1) and Norfolk Ridge (3) domains, it accounts for roughly 4 %. Outside and with increasing distance to the generation sites, i.e., in the far-field, $F_{\mathrm{bc}}^{\mathrm{inc}}$ becomes more important with ratios $F_{\mathrm{bc}}^{\mathrm{inc}}/F_{\mathrm{bc}}^{\mathrm{D2}} =$ 0.5-0.9 (not shown). However, it is important to note that in these cases $F_{\mathrm{bc}}^{\mathrm{D2}}$ is considerably reduced ($<5$ kW m$^{-1}$).

The residual $D_{\mathrm{bc}}^{\mathrm{D2}}$ is here taken as a proxy for energy dissipation following the discussion in Sect. 2.2. Integrated over the subdomains North (1), South (2), Norfolk Ridge (3), and Loyalty Ridge (4), 38 %, 58 %, 40 %, 29 % of the locally generated energy are dissipated in the near-field, respectively (Fig. 1g, Table 1). $D_{\mathrm{bc}}^{\mathrm{coh}}$ is representative of both coherent energy dissipation and energy being removed from the coherent internal tide through non-linear energy transfers (Fig. 1h). As expected, $D_{\mathrm{bc}}^{\mathrm{coh}}$ is generally increased compared to $D_{\mathrm{bc}}^{\mathrm{D2}}$. The difference between both ($D_{\mathrm{bc}}^{\mathrm{inc}}$) can be understood as the fraction by which energy dissipation is overestimated in $D_{\mathrm{bc}}^{\mathrm{coh}}$ or the energy gained by the incoherent tide. This amounts to 10 %, 9 %, and 22 % in the South (2), Norfolk Ridge (3), and Loyalty Ridge (4) domains, respectively (Fig. 1i and Table 1). In the North (1) domain, it is slightly positive (3 %) suggesting net energy dissipation linked to the incoherent tide. This is a worth noting conclusion for Part 1, suggesting that the coherent dissipation rates for the largely dominating M2 tidal constituent may be overestimated by $\sim$10-20 %.





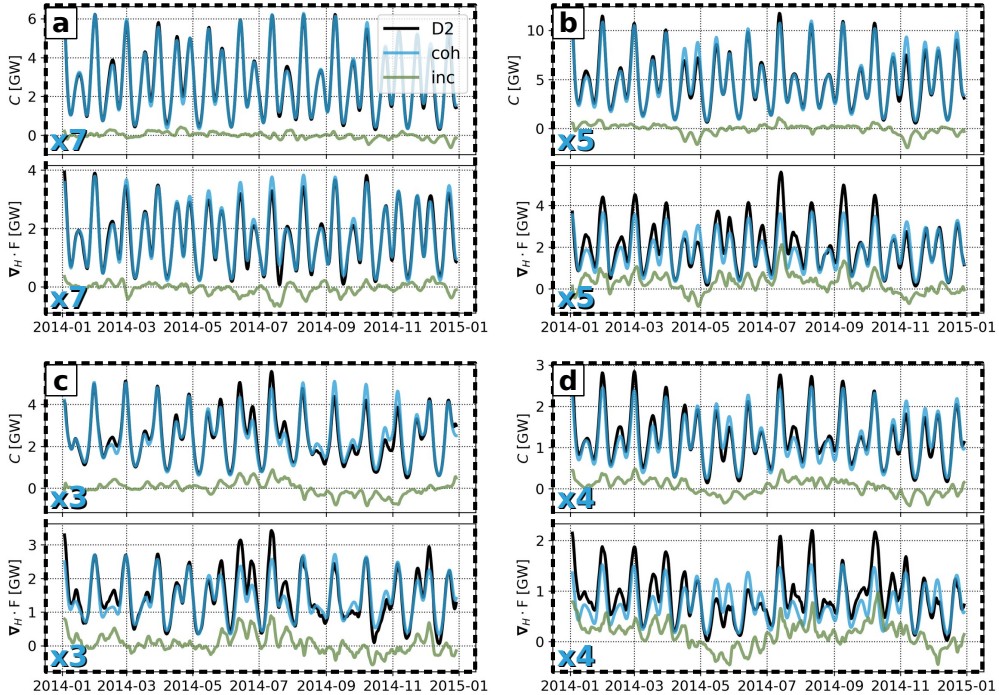

**Figure 2.** Time series of semidiurnal (black) barotropic-to-baroclinic conversion $C^{D2}$ and energy flux divergence $\nabla_h \cdot \mathbf{F}_{bc}^{D2}$, decomposed into the coherent (blue) and incoherent (green) components, integrated over (a) North, (b) South, (c) Norfolk Ridge, (d) Loyalty Ridge. The mean factor of change for $C^{coh}$ and $\nabla_h \cdot \mathbf{F}_{bc}^{coh}$ between spring and neap tide ($\sim$2 weeks) is also given (in blue in the lower left corner).

## 4 Semidiurnal barotropic-to-baroclinic energy conversion variability

In the annual mean, the semidiurnal barotropic-to-baroclinic conversion is largely dominated by the coherent component. In the meanwhile, it is the coherent tide which is associated with a substantial amount of variability due to the astronomically forced fortnightly modulated spring-neap cycle, i.e., the interaction of M2 and S2 tidal constituents (Fig. 2). Note that the N2 tidal constituent adds a low-frequency component to the modeled variability with a period of $\sim$9 months. Specifically, on time scales of $\sim$2 weeks conversion linked to the coherent tide may vary on average by a factor which ranges between 3 and 7 among the subdomains between spring and neap tides. This is in agreement with recent findings over the Reykjanes Ridge (Vic et al., 2021). By doing so, it explains in the area-integral most of the semidiurnal conversion variability ($\gamma^{coh} = 90\text{-}99$ %) within the internal-tide generation hot spots (Table 2). The remaining fraction is explained by the incoherent tide ($\gamma^{inc} = 1\text{-}10$ %).

We explicitly show the spatial contribution to semidiurnal conversion variability by the coherent and incoherent parts for the South (2) domain (Fig. 3). The South (2) domain represents the most prominent internal-tide generation site with conversion rates well above 1 W m$^{-2}$ across the steep slopes such as Pines Ridge (Fig. 3a). Even though, it is the coherent tide and, thus, the spring-neap cycle which dominates semidiurnal variability, certain regions stand out by increasing contributions of the




**Table 2.** Domain-averaged explained variability of the coherent ($\gamma^{\mathrm{coh}} = \mathrm{cov}(C^{\mathrm{coh}}, C^{\mathrm{D2}})/\mathrm{var}(C^{\mathrm{D2}})$)) and incoherent ($\gamma^{\mathrm{inc}}$) barotropic-to-baroclinic conversion referenced to to the semidiurnal conversion variability ($C^{\mathrm{D2}}$). The domain-averaged explained variability of the purely incoherent term $\gamma^{\mathrm{inc}^*} = \mathrm{cov}(C^{\mathrm{inc}^*}, C^{\mathrm{inc}})/\mathrm{var}(C^{\mathrm{inc}})$)) and the two cross-terms ($\gamma^{\mathrm{cross1}}, \gamma^{\mathrm{cross2}}$) relative to the incoherent conversion variability ($C^{\mathrm{inc}}$) is also given. We refer to Equation 6 for the decomposition of $C^{\mathrm{D2}}$ and $C^{\mathrm{inc}}$.

|  | North | South | Norfolk Ridge | Loyalty Ridge |
|---|---|---|---|---|
| $\gamma^{\mathrm{coh}}$ | 0.98 | 0.99 | 0.97 | 0.90 |
| $\gamma^{\mathrm{inc}}$ | 0.02 | 0.01 | 0.03 | 0.10 |
| $\gamma^{\mathrm{inc}^*}$ | 0.04 | 0.00 | -0.01 | 0.04 |
| $\gamma^{\mathrm{cross1}}$ | 0.73 | 0.91 | 1.06 | 0.89 |
| $\gamma^{\mathrm{cross2}}$ | 0.23 | 0.09 | -0.05 | 0.07 |

incoherent tide ($C^{\mathrm{inc}}$). This is the case close to the lagoon and to the southeast around seamounts, notably Antigonia, Jumeaux Est, Jumeaux Ouest, and Stylaster (Fig. 3c). In these regions, the incoherent tide can explain well above 50 % of the semidiurnal conversion variability. Though, it is worth mentioning that they are generally associated with reduced values of $C^{\mathrm{D2}}$ compared to the overwhelmingly strong generation at Pines Ridge. This is crucial knowledge for the design and interpretation of in-situ observations at fixed locations such as moorings (see Sect. 7).

To better understand the origin of $C^{\mathrm{inc}}$, we further decompose it into a purely incoherent term ($C^{\mathrm{inc}^*}$) and two cross-terms ($C^{\mathrm{cross1}}, C^{\mathrm{cross2}}$) following Equation 6. Across all internal-tide generation hot spots, $C^{\mathrm{cross1}}$ dominates ($\gamma^{\mathrm{cross1}} > 73$ %; Table 2). This implies that semidiurnal conversion variability (apart from the tidal-forcing induced spring-neap variability) is driven by the work of the barotropic tide on baroclinic bottom pressure variations $p_{\mathrm{bc}}^{\mathrm{inc}}(-H)$. $C^{\mathrm{cross2}}$, linked to temporal variations

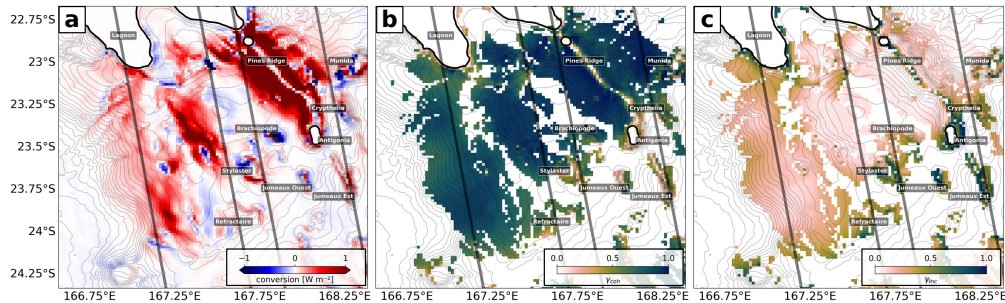

**Figure 3.** (a) Annual-mean, depth-integrated, semidiurnal barotropic-to-baroclinic energy conversion as in Fig. 1a, but zoomed into the South (2) domain. Explained variability by the (b) coherent ($\gamma^{\mathrm{coh}} = \mathrm{cov}(C^{\mathrm{coh}}, C^{\mathrm{D2}})/\mathrm{var}(C^{\mathrm{D2}})$) and (c) incoherent ($\gamma^{\mathrm{inc}} = \mathrm{cov}(C^{\mathrm{inc}}, C^{\mathrm{D2}})/\mathrm{var}(C^{\mathrm{D2}})$) component. Explained variability is only shown for regions, where $|C^{\mathrm{D2}}| > 0.05$ W m$^{-2}$. The depth contour interval is 100 m. The SWOT swaths (solid black lines) during the fast sampling phase (1 d repeat orbit) are also shown.





of the barotropic forcing, can account for up to $\gamma^{\mathrm{cross2}} = 23$ %. $C^{\mathrm{inc}^*}$ tends to play a negligible role. We note that $C^{\mathrm{inc}^*}$ and $C^{\mathrm{cross2}}$ can have compensating effects (see Norfolk Ridge (3) in Table 2), but the mechanisms underlying this compensation remain unclear.

Similarly to the above, we show for the South (2) domain the contribution of the different terms, which make up $C^{\mathrm{inc}}$ (Fig. 4). The annual mean for $C^{\mathrm{inc}^*}$, $C^{\mathrm{cross1}}$, and $C^{\mathrm{cross2}}$ are shown in Fig. 4a-c. Note that the colorbar range is an order of magnitude smaller than in Fig. 1a-c, and that the overall contribution in the annual mean remains small. The three terms feature similar amplitudes. Though, the spatial patterns differ from each other. Corresponding to the area-integrated explained variability in Table 2 ($\gamma^{\mathrm{cross1}} = 91$ %), and alongside high standard deviations, $C^{\mathrm{cross1}}$ dominates $C^{\mathrm{inc}}$ (Fig. 4e,h). $C^{\mathrm{cross2}}$ plays a minor, but

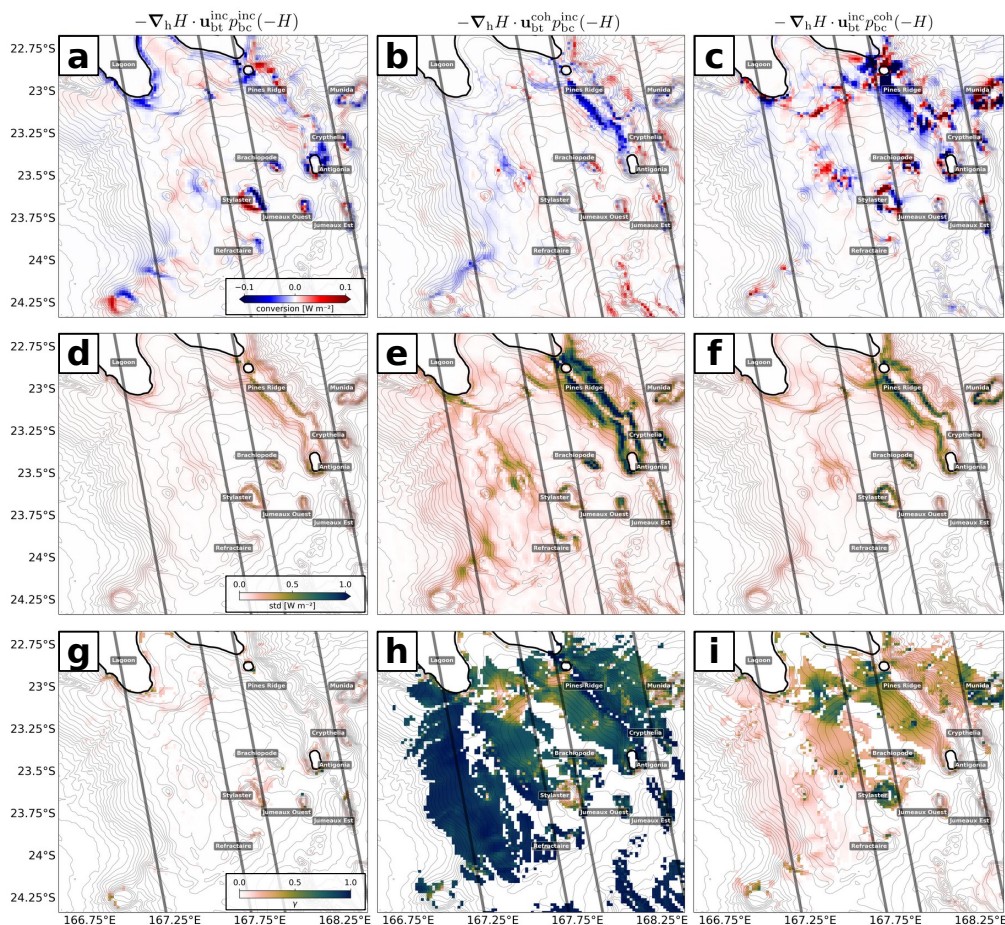

**Figure 4.** Annual mean and standard deviation of depth-integrated semidiurnal barotropic-to-baroclinic incoherent energy conversion separated into the (a,d) purely incoherent ($C^{\mathrm{inc}^*}$) term, (b,e) first ($C^{\mathrm{cross1}}$) and (c,f) second ($C^{\mathrm{cross2}}$) cross-terms. The explained variability of (g) $C^{\mathrm{inc}^*}$, (h) $C^{\mathrm{cross1}}$, and (i) $C^{\mathrm{cross2}}$ are referenced to the incoherent conversion $C^{\mathrm{inc}}$.





non-negligible role ($\gamma^{\mathrm{cross1}} = 9$ % of the incoherent variance) with elevated contribution particularly above steep bathymetry (Fig. 4e,h).

To summarize, incoherent conversion averages approximately to zero in the annual mean. Temporally, though, it features positive and negative anomalies associated with enhanced and reduced semidiurnal conversion, respectively (see Fig. 2). Decomposing $C^{\mathrm{inc}}$ allows for a more detailed view on the origin of conversion variations not linked to spring-neap tide variability. As expected, the work of the barotropic tide on baroclinic bottom pressure variations $p_{\mathrm{bc}}^{\mathrm{inc}}(-H)$, expressed by $C^{\mathrm{cross1}}$ is the most important. To our surprise, conversion variability induced by temporal variations of the barotropic forcing ($C^{\mathrm{cross2}}$) is non-negligible. Temporal variations of the barotropic tide are generally known to exist through seasonal (Müller et al., 2012; Yan et al., 2020) and climatological (Opel et al., 2024) stratification changes. It may also be possible that these temporal variations represent the energy transfer from the internal tide to the barotropic tide due to pressure work (Zilberman et al., 2009). Yet, to our knowledge it remains to be quantified to what extent they may drive conversion variability. Further effort is needed in this regard. Also, we can not fully exclude uncertainties linked to the applied methodology, i.e., the bandpass filter and depth-average to extract the semidiurnal barotropic tide. In the following, we put the focus on the governing processes, which drive baroclinic bottom pressure variations $p_{\mathrm{bc}}^{\mathrm{inc}}(-H)$.

## 4.1 Mesoscale-eddy-induced conversion variations

Generally speaking, $C^{\mathrm{cross1}}$ is driven by baroclinic bottom pressure variations, which can be due to local and remote effects through local stratification changes and shoaling of remotely-generated internal tides, respectively. The former express by pressure amplitude variations ($dP_A$), whereas the latter express by pressure phase variations ($dP_\phi$). We start by focusing on Pines Ridge in the South (2) domain before generalizing our findings (Fig. 5). The monthly time series of conversion variations expressed by the ratio of $C^{\mathrm{cross1}}/C^{\mathrm{D2}}$ reveals two distinct events around April/May and November 2014, during which conversion is suppressed by more than 10 %. Here, conversion variability through $C^{\mathrm{cross1}}$ is largely driven by baroclinic bottom pressure amplitude variations (correlation coefficient r=0.80 with a 90 % confidence interval [0.50, 0.93] assuming N=12 samples) (Fig. 5a). Moreover, there is no correlation with baroclinic bottom pressure phase variations (r=-0.02 [-0.51,0.49]). $C^{\mathrm{cross1}}$ does not follow a seasonal cycle. Rather, the modeled variability on monthly to intraseasonal time scales are highly suggestive of mesoscale variability.

Following Kunze et al. (2002), stratification and pressure perturbations are linked via $p(z) = \int_z^0 N^2 \eta \, dz' - (1/H) \int_{-H}^0 \int_z^0 N^2 \eta \, dz' dz$, where $\eta$ is the vertical displacement of isopycnal surfaces. Here, bottom stratification $N^2(-H)$ is extracted from the bottom most grid cell. The negative conversion/bottom pressure amplitude anomalies in Fig. 5b are associated with negative bottom stratification anomalies and negative mesoscale SLA, alongside elevated mesoscale EKE (Fig. 5b). Mesoscale SLA, surface geostrophic velocity, and surface mesoscale EKE are computed similarly to Sect. 3.2 in Part 1. Briefly, we computed 5 d mean SSH to eliminate high-frequency variability before binning the data onto a 1/4°horizontal grid. Finally, we applied a high-pass



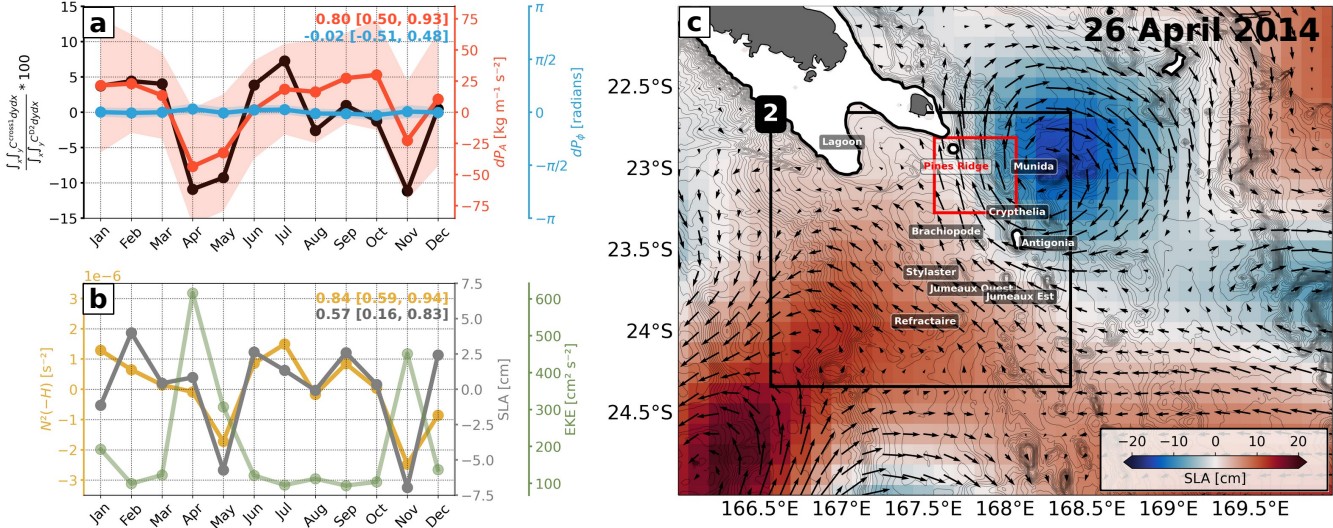

**Figure 5.** (a) Monthly time series of conversion anomaly (black) integrated over Pines Ridge (red box in (c)), expressed by the ratio $C^{\mathrm{cross1}}/C^{\mathrm{D2}}$. Also shown are the domain-averaged baroclinic bottom pressure amplitude ($dP_A$, red) and phase ($dP_\phi$, blue) difference between the semidiurnal and coherent tide (including standard deviation). (b) Monthly time series of the domain-averaged bottom stratification ($N^2(-H)$, yellow), mesoscale SLA (gray) and EKE (green). The correlation coefficients of (a) $C^{\mathrm{cross1}}$ with $dP_A$, $dP_\phi$ and of (b) $dP_A$ with $N^2(-H)$, mesoscale SLA including their 90 % confidence intervals are also given. (c) 5-day mean of mesoscale SLA including the geostrophic velocity field reveals a cyclonic eddy approaching Pines Ridge (red box).

filter with the region's characteristic cut-off period of 180 d to account for the mesoscale. The 5-day mean of mesoscale SLA
including the surface geostrophic velocity field, exemplarily shown for 26 April, reveals a cyclonic eddy approaching Pines Ridge (Fig. 5c).

Assuming that mesoscale eddies with strong surface signature have a depth-independent structure, i.e., barotropic or weakly baroclinic, we suggest that conversion and mesoscale variability are linked through the upward and downward pumping of
isopycnals induced by cyclonic (CE) and anticyclonic (AE) eddy activity, respectively. In turn, the upward and downward pumping favors decreasing (increasing) stratification above the seafloor and, thus, equivalent variations in baroclinic bottom pressure and conversion (see also Fig. A1). In phase with the local tidal forcing, $p_{\mathrm{bc}}^{\mathrm{inc}}(-H)$ induced by AE adds constructively to $p_{\mathrm{bc}}^{\mathrm{coh}}(-H)$, whereas $p_{\mathrm{bc}}^{\mathrm{inc}}(-H)$ induced by CE is in opposite phase and has a destructive effect. This is supported by positive correlations of pressure amplitude variations with bottom stratification (r=0.84 [0.59, 0.94]) and mesoscale SLA (r=0.57 [0.10,
0.83]) in Fig. 5b.

Generalizing our results for the whole region is far from being straightforward. We calculate the probability distribution of the correlation coefficients for the monthly time series of $C^{\mathrm{cross1}}$ with $dP_A$, $dP_\phi$, $N^2(-H)$, and mesoscale SLA for each





subdomain (Fig. 6). The correlation coefficients for the domain-integrated/averaged quantities (filled circles) including their
90 % confidence intervals are also given. Generally, pressure amplitude variations are very pronounced suggestive of local
effects (Fig. 6a-d). Correlations with pressure phase variations tend to be less pronounced or more randomly distributed. Ex-
empt therefrom is Norfolk Ridge (3), where pressure phase variations are strongly positively correlated suggesting that remote
effects are important (Fig. 6c). Note that pure correlations do not provide information about the amplitude of $C^{\text{cross1}}$. High
correlations indicate that the temporal patterns of variability are consistent, but they do not imply that the magnitude of the
conversion is equally significant across all regions.

Assuming that local effects dominate overall, the probability distribution for the correlation of pressure amplitude variations
with bottom stratification and mesoscale SLA are shown in (Fig. 6e-h). In Fig. 6f and g (representative of the South (2) and Nor-
folk Ridge (3) domains), the slightly negatively skewed probability distributions are statistically robust with the hypothesis that

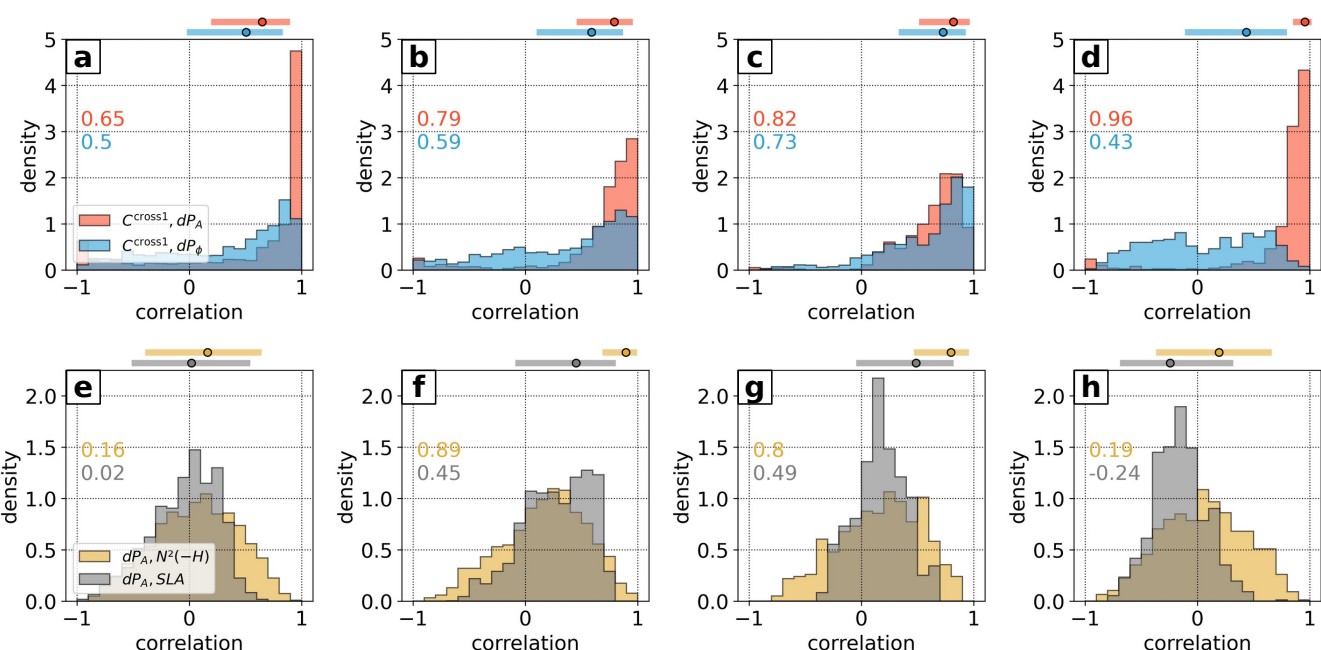

**Figure 6.** Histograms of the correlation coefficient between monthly averaged $C^{\text{cross1}}$ and baroclinic bottom pressure amplitude ($dP_A$, red) and phase ($dP_\phi$, blue) differences between the semidiurnal and coherent tide, determined by complex demodulation of $p_{\text{bc}}^{\text{D2}}(-H)$ and $p_{\text{bc}}^{\text{coh}}(-H)$ for (a) North, (b) South, (c) Norfolk Ridge, (d) Loyalty Ridge. We consider only regions where $|C^{\text{D2}}| > 0.2$ W m$^{-2}$ to emphasize areas of strong conversion. (e-h) Same as above but for the correlation between $dP_A$ and bottom stratification ($N^2(-H)$, yellow) and mesoscale SLA (gray). The correlation coefficients for the domain-integrated/averaged quantities (filled circles) including their 90 % confidence intervals are also given above the panels. The associated time series are explicitly shown in Fig. A2. Note that correlation coefficients based on the domain-integrated/averaged time series can largely differ from the probability distributions, as explained in Sect. 4.1.





conversion variations are linked with mesoscale-eddy-induced stratification changes. This becomes more evident, when considering the correlation coefficients based on the area-integrated/averaged time series (see the filled circles above the panels). There are good correlations with bottom stratification (South (2): r=0.89 [0.71,0.96]; Norfolk Ridge (3): r=0.80 [0.49,0.93]) and mesoscale SLA (South (2): r=0.45 [-0.06,0.78]; Norfolk Ridge (3): r=0.49 [-0.02, 0.79]). In these regions, mesoscale-eddy-induced stratification changes can induce semidiurnal conversion anomalies by up to 20 % (see Fig. A2). These correlations

(based on the area-integrated/averaged time series) can largely differ from the regional probability distributions, as the former may give greater weight to regions of enhanced conversion influenced by mesoscale variability. In contrast, the probability distributions represent the correlations at every grid point. We only choose grid points, where $|C^{\mathrm{D2}}| > 0.2$ W m$^{-2}$ to emphasize areas of strong conversion. We note that as the threshold increases, the probability distributions become more negatively skewed, indicating increasingly positive correlations of $dP_A$ with $N^2(-H)$ and mesoscale SLA (not shown).


North (1) and Loyalty Ridge (4) show no clear correlations (Fig. 6e,h). Several explanations are possible. First, the assumption of depth-independent or weakly baroclinic structure may not be valid. Particularly, conversion at Loyalty Ridge (4) takes place in deeper waters (1000-3000 m compared to <1000 m in South (2)) and, thus, considerably deeper than the eddies' vertical extent. Mesoscale eddies near North (1) are generally less numerous and much less energetic than south of New Caledonia

(Keppler et al., 2018). There are many other factors, which can alter conversion such as seasonal stratification changes. Seasonal stratification changes have recently been reported to drive conversion variations on global scales (Kaur et al., 2024). In our study region, they seem to play a secondary role and are at best superimposed on mesoscale variability (see Fig. A2). Having said that, remote effects can play an essential role in local conversion variations, too. They include primarily the shoaling of remotely generated internal tides, which undergo phase modulations as they propagate through the open ocean before

impinging on bathymetric slopes, which are subject to local internal-tide generation. They are usually out of phase with the local tide forcing. Though, they can theoretically be in phase or in opposite phase as well, making it hard to distinguish local from remote effects. However, remote effects seem to be of relatively small importance around New Caledonia except at minor internal-tide generation sites, which lie in propagation direction of the major tidal beams. Moreover, there are no major remote sources of internal tides, which potentially shoal on the New Caledonia ridges.


We conclude that mesoscale variability can be an important source of conversion variations by enhancing/reducing semidiurnal energy conversion within the internal-tide generation hot spots around New Caledonia. Generalizing our findings is challenging since the relative importance of the underlying dynamics can strongly vary among the generation sites. Furthermore, various different processes may be superimposed. Nonetheless, on monthly to intraseasonal time scales we attribute 10-20 % of semid-

iurnal conversion variations to mesoscale-eddy-induced stratification changes.





## 5 Tidal incoherence in the far-field

Once generated, semidiurnal internal tides propagate in narrow tidal beams towards the open ocean. Within the generation hot spots, the variability of semidiurnal energy flux divergence is closely correlated to the one of semidiurnal conversion and, thus,

follows the spring-neap tide cycle (Fig. 2). In the annual mean, tidal incoherence of energy flux divergence can account for an important fraction, which stands in contrast to the conversion term (see Sect. 3, Table 1). We will show in the following that tidal incoherence becomes increasingly important in the far-field, linked to mesoscale eddy variability around New Caledonia. First evidence is given by the 5-day mean of $F_{bc}^{inc}$ and the surface geostrophic velocity field (Fig. 7a). In the influence area or in propagation direction of the tidal beams, elevated incoherent energy levels $>10\,\mathrm{kW\,m^{-1}}$ are clearly associated with intensified

mesoscale currents. This suggests that the semidiurnal energy flux becomes incoherent as it propagates through the eddy field, especially south of New Caledonia where mesoscale EKE is enhanced (Fig. 7b). Reduced levels of incoherent energy fluxes north of New Caledonia correspond with weaker mesoscale surface currents.

### 5.1 Mesoscale-eddy-induced refraction of tidal beam energy propagation

To better understand the underlying mechanism, we applied a simplified ray tracing following Rainville and Pinkel (2006) and

Bendinger et al. (2024) (see Sect. 2.3). The primary objective is to study the refraction of tidal beam energy propagation by

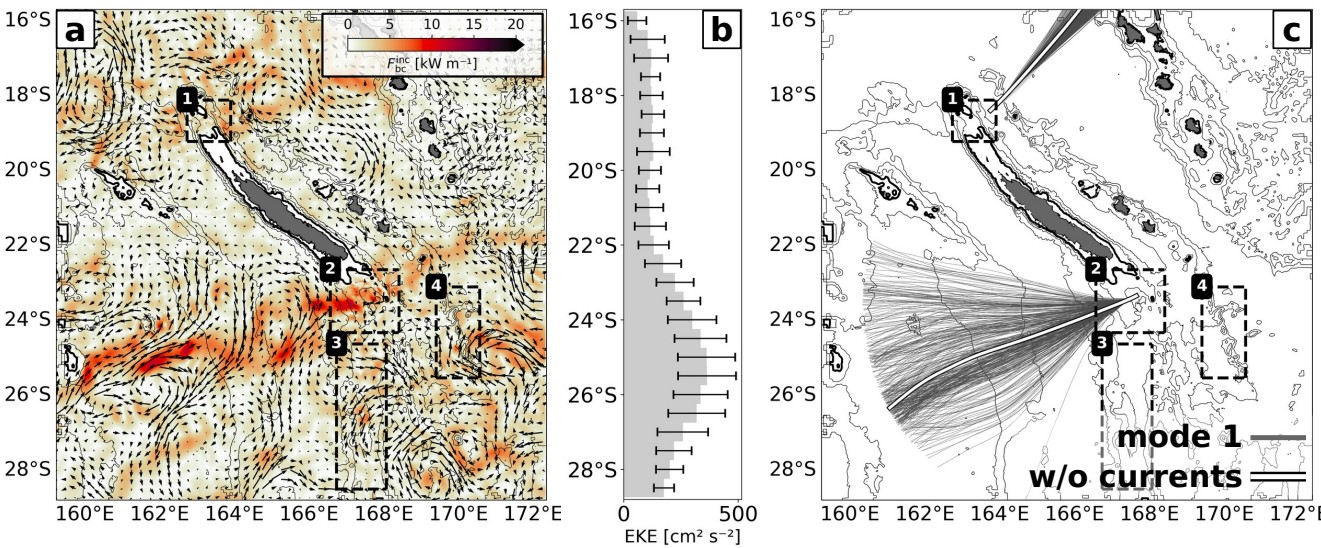

**Figure 7.** (a) 5-day mean (31 January 2014, spring tide) of incoherent energy flux ($F_{bc}^{inc}$) overlaid by the surface geostrophic velocity field representative of the mesoscale eddy field. (b) Zonally averaged mesoscale EKE (including standard deviation) as a function of latitude. (c) Modeled semidiurnal ray paths for mode 1 propagating through the mesoscale eddy field, from their initialization north (163.75° E, 18.4° S) and south (167.65° E, 23.35° S) of New Caledonia based on daily-mean, depth-averaged background currents (model year 2014). The no-currents scenario is also given (highlighted white line).



**Table 3.** Semidiurnal ray's mean group arrival time (after 500 iterations, equivalent to 500 km distance given a step size of 1 km) for modes 1-2 without (w/o) and with (w/) currents initiated from the North (1) and South (2) domains. The average group arrival time delay relative to the no-currents scenario is also given.

| North (1) | mode 1 | mode 2 |
|---|---|---|
| w/o currents [days] | 2.29 | 4.38 |
| w/ currents [days] | $2.38 \pm 0.08$ | $4.7 \pm 0.26$ |
| delay [hours] | $2.18 \pm 1.82$ | $7.8 \pm 6.23$ |
| South (2) | mode 1 | mode 2 |
| w/o currents [days] | 2.68 | 5.40 |
| w/ currents [days] | $2.65 \pm 0.06$ | $5.25 \pm 0.22$ |
| delay [hours] | $-0.86 \pm 1.44$ | $-3.64 \pm 5.2$ |

considering only effects of depth-independent currents. To do so, we initiated a semidiurnal ray for two initialization regions in the North (1) and South (2) domains. The theoretical mode-1 ray paths for daily-mean, depth-averaged background currents are shown in Fig. 7c. The no-current scenario is also shown (w/o currents). The ray tracing yields profoundly different results for the tidal beam energy propagation north and south of New Caledonia. Theoretical rays initiated in the North (1) domain are

confined in well-defined and narrow tidal beams, which closely align with the theoretical propagation direction for a semidiurnal ray in the absence of background currents. This stands in contrast to the theoretical rays initiated in the South (2) domain, experiencing notable refraction in propagation direction. For both regions, mode 2 is more affected by the background currents (not shown).

Here, we associate the tidal beam's refraction with elevated mesoscale EKE (see Fig. 7b). Mesoscale activity is clearly elevated south of New Caledonia, explaining the contrast between the North (1) and South (2) domains and to what extent a theoretical ray is being refracted. In agreement with Keppler et al. (2018), we find that mesoscale eddies south of New Caledonia are more long-lived and more energetic as they feature higher rotational speeds than those north of New Caledonia (not shown). The physical mechanism lies in the background-currents-induced changes of the wave's group and phase speeds, which in

turn cause the tidal beam to refract or change orientation, alongside increasing phase variability or phase offset in propagation direction relative to the no-currents scenario. Overall, there is increasing dispersion south of New Caledonia representing the stochastic nature of the mesoscale eddy field. Changing group speeds express inter alia in the wave's group arrival time at a given location (Table 3). The mean group arrival time for mode 1 initiated from South (2) is 2.65±0.06 d, corresponding with a delay of -0.86±1.44 d relative to the arrival time in the no-currents scenario. Mode 2 is substantially more delayed than mode

1, roughly by a factor 4 or larger, in good agreement with the findings in Rainville and Pinkel (2006). Corresponding with the mean current direction, the theoretical rays initiated from the South (2) domain reach the far-field earlier than expected in the no-currents scenario. In contrast, the theoretical rays initiated from the North (1) domain reach the far-field later than expected.



It is worth mentioning that mesoscale eddies can be associated with stratification anomalies, which alter ray propagation.
Keppler et al. (2018) reported strong anomalies in temperature and salinity within mesoscale eddies down to 1000 m depth south of New Caledonia. It remains unclear to what extent those stratification anomalies impact the propagation of semidiurnal internal tides in our region. Nonetheless, from idealized simulations it is suggested that the predominant contribution to tidal beam refraction lies in background currents, whereas stratification anomalies associated with mesoscale eddies only play secondary role (Guo et al., 2023).

## 6 Implications of tidal incoherence for SSH observability of balanced and unbalanced motions

Internal tides typically manifest in SSH as variations on the order of a few centimeters. Conventional satellite altimetry has taken advantage of this to derive robust global estimates of the coherent internal-tide SSH signature using over 20 years of data. In Part 1, the SSH signature of the coherent M2 internal tide was analyzed, revealing multiple interference patterns as M2 tidal waves emanated from several generation sites, interacting constructively and destructively along their propagation paths. The results showed good agreement with empirical estimates from satellite altimetry (HRET; Zaron, 2019), with reasonable amplitudes and spatial patterns for both the dominant mode 1 (up to 6 cm) and mode 2 (up to 2 cm).

It remains to be investigated how the incoherent internal tides manifests in SSH, particularly regarding the implications of tidal incoherence, i.e., eddy-internal tide interactions, for the observability of balanced and unbalanced motions. These insights may provide crucial information for the dynamical interpretation of SSH measurements from satellite altimeter mission such as SWOT. Disentangling these dynamics represents a major challenge since SSH may contain contributions from both mesoscale to submesoscale motions and internal gravity waves. Furthermore, these dynamics can have comparable temporal and spatial scales and contribute equally to SSH variance. The New Caledonia region provides an excellent study site, characterized by strong mesoscale variability and energetic internal tides.

The transition scale, $L_\mathrm{t}$, provides a quantitative measure to distinguish between balanced and unbalanced motions (Qiu et al., 2018; Vergara et al., 2023). It represents the length scale in spectral space where unbalanced motions begin to dominate over balanced motions. Transition scales are commonly derived from one-dimensional SSH wavenumber spectra along altimeter tracks or two-dimensional spectra from numerical simulations, revealing strong geographic and seasonal variability. For instance, submesoscale processes are typically more energetic in winter months (Callies et al., 2015; Rocha et al., 2016), while internal tides often feature amplified SSH signatures in summer due to enhanced surface stratification (Lahaye et al., 2020; Kaur et al., 2024). However, the underlying methodologies have inherent limitations, particularly the assumption of isotropy. While isotropy may be a reasonable approximation for mesoscale and submesoscale circulation, it does not apply to processes such as internal tides, which propagate in well-defined directions. As a result, this assumption can lead to biased interpretations because the methodology emphasizes isotropic motions.





In the following, we investigate the SSH imprint of the semidiurnal incoherent tide. Implications of tidal incoherence for SSH observability of balanced and unbalanced motions are deduced by computing SSH wavenumber spectra both in the direction of tidal energy propagation and along realistic altimetry tracks, we explore the implications of tidal incoherence for the observability of balanced and unbalanced motions around New Caledonia.

## 6.1 Semidiurnal SSH decomposed into its coherent and incoherent parts

We extend the analysis in Sect. 2.2 by decomposing semidiurnal SSH into its coherent and incoherent parts (Fig. 8). The overall signature resembles the semidiurnal energy flux in Fig. 1d-f with the predominant beams to the north and south of New Caledonia, clearly visible in SSH with root-mean-squares (rms) of $> 6$ cm and dominated by the coherent tide (Fig. 8a-b). The SSH manifestation of the incoherent tide is characterized by an overall smaller but important rms (1-2 cm; Fig. 8c). The incoherent SSH is less confined to the tidal beams and seems more widespread in the domain, which suggests that the dispersion of internal waves occurs all along their propagation through the domain.

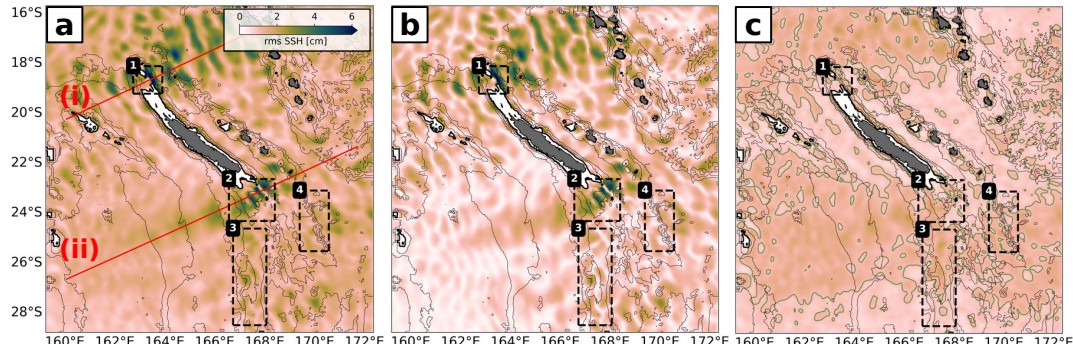

**Figure 8.** Annual root-mean-square (rms) of (a) semidiurnal SSH decomposed into the (b) coherent and (c) incoherent components. For the incoherent SSH, the 1 cm contour is shown (green). Bathymetry contours and the black boxes are given as in Fig. 1. (i) and (ii) in (a) are the transects in tidal energy propagation direction, which are used to compute SSH wavenumber spectra (see Sect. 6.3).

## 6.2 SSH wavenumber spectra in tidal beam energy propagation direction: an optimal case study

In Part 1, we calculated annually averaged SSH wavenumber spectra along the beam direction for two transects: (i) a northern transect and (ii) a southern transect (see Fig. 8a). The objective was to investigate the underlying dynamical regimes and assess the relative importance of balanced and unbalanced motions. The analysis revealed that the coherent internal tide strongly dominates SSH variance within the mesoscale band (70–250 km). The estimated transition scales for the northern and southern transects were estimated 204 km and 163 km, respectively, suggesting that SSH observability of balanced motions is primarily limited to large eddy scales. Applying a correction for the coherent internal tide significantly reduced the transition scales ($L_t^{\mathrm{corr}}$) to 92 km and 83 km, respectively.





Here, we revisit this analysis by addressing the incoherent internal-tide to understand its impact on transition scales and potentially for SSH observability of balanced motions. Similarly to Part 1, we investigate SSH wavenumber spectra with regard to different dynamics that are separated in terms of frequency bands: subinertial frequencies ($\omega < f$, $\mathrm{SSH}_{\mathrm{subinertial}}$) for

mesoscale and submesoscale dynamics, as well as superinertial frequencies ($\omega > f$, $\mathrm{SSH}_{\mathrm{superinertial}}$) for internal gravity waves while distinguishing between the coherent internal-tide ($\mathrm{SSH}_{\mathrm{coh}}$), incoherent internal-tide ($\mathrm{SSH}_{\mathrm{inc}}$), and supertidal frequencies ($\omega >$1/10 h, $\mathrm{SSH}_{\mathrm{supertidal}}$). Note that the incoherent internal-tide is determined by applying a bandpass filter in the full semidiurnal-diurnal tidal range (10-28 h). As such, it also comprises contributions from near-inertial, non-tidal internal gravity waves, short-live submesoscale features, but we assume that they are negligibly small.


Seasonally averaged SSH spectra for Southern Hemisphere summer (January-March, JFM) and winter (July-September, JAS) are shown in Fig. 9. By definition, the coherent internal tide is the same in both seasons. The seasonality of $L_{\mathrm{t}}^{\mathrm{corr}}$ is, thus, generally attributed to seasonal variations of subinertial motions, i.e., mesoscale to submesoscale dynamics, and unbalanced wave motions. First, we point out that the incoherent internal-tide ($\mathrm{SSH}_{\mathrm{inc}}$) predominantly governs motions at superinertial

frequencies if corrected for the coherent internal-tide - at least down to 100 km independent of the season for both transects.

Along the southern transect, seasonal modulations of the SSH spectra become evident for all wavelengths < 300 km (Fig. 9c-d). In summer, it features a more flattened wavenumber slope in the mesoscale to submesoscale range with a characteristic $k^{-2}$ slope corresponding to superinertial motions (internal wave continuum) (Fig. 9c). In winter, it becomes more continuous,

characterized by a $k^{-4}$ slope (Fig. 9d). This can be attributed to subinertial motions such as mesoscale to submesoscale processes that undergo seasonal variability. This was explicitly shown for New Caledonia in Sérazin et al. (2020) and Bendinger (2023), in which increasing importance of mixed layer instabilities and frontogenesis was attributed to more available potential energy in the Southern Hemisphere winter months. Seasonal modulations of SSH spectra can also be linked to unbalanced wave motions which are amplified in summer months, particularly for higher-vertical modes due to increasing stratification

(Lahaye et al., 2020; Kaur et al., 2024). Superinertial processes dominate subinertial motions in both seasons at scales below 180 km. However, the relative importance of superinertial over subinertial motions is more pronounced in summer months. This can be explained by the seasonality of superinertial and subinertial motions being out of phase, i.e., superinertial motions are enhanced in summer while subinertial motions are considerably reduced and vice versa in winter. The transition scale ($L_{\mathrm{t}}$) does not feature strong seasonality between summer and winter (Fig. 9c,d). Though, the transition scale is not well defined in

winter, where subinertial and superinertial signals have similar variance at wavelengths 90-180 km (Fig. 9d).

Important conclusions are made when correcting for the coherent internal tide. In summer, the transition scale ($L_{\mathrm{t}}^{\mathrm{corr}}$) is only slightly reduced from 175 km to 156 km (Fig. 9c). This is linked to the seasonally enhanced incoherent internal tide which is still contained in the signal featuring equal SSH variance with subinertial signals at 90-180 km wavelength (Fig. 9c). At

scales below 90 km, the incoherent tide even dominates SSH variance over subinertial motions and is equally important to the coherent tide. In winter, the transition scale is largely reduced from 165 km to 78 km (Fig. 9d). This is primarily linked to the

logo

 

fact that subinertial motions are energized in winter while the SSH signature of motions at superinertial frequencies are less energetic as stated above. We note a significant contribution of motions at supertidal frequencies for scales smaller than 100 km.

The northern domain differs from the southern domain in that SSH variance of subinertial motions is generally reduced and the seasonal cycle less pronounced (shown by the white curves in Fig. 9a-b). Motions at superinertial frequencies largely govern over motions at subinertial frequencies throughout the year, dominated by the coherent internal tide. As for the southern tran-

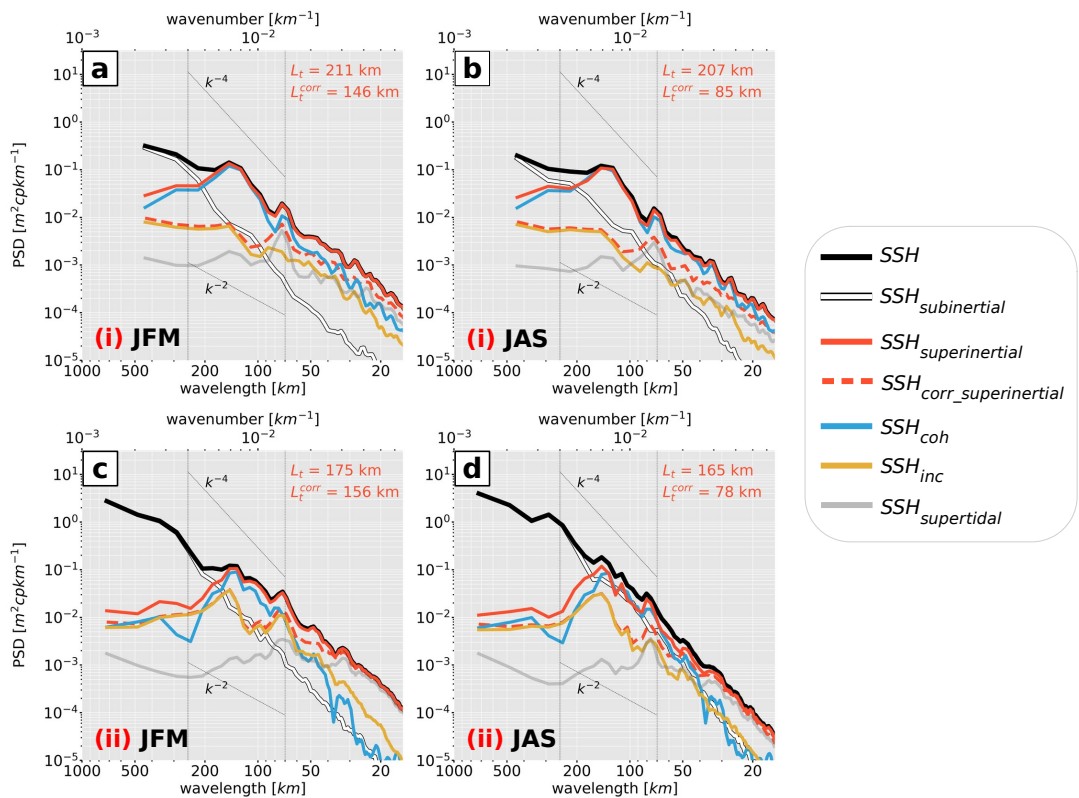

**Figure 9.** Seasonally averaged SSH wavenumber spectra, i.e., Southern Hemisphere summer (January-March, JFM) and winter (July-September, JAS), for transects **(a)-(b)** north and **(c)-(d)** south of New Caledonia, denoted (i) and (ii), respectively, in Fig. 8a. SSH spectra are presented for the altimetry-like SSH (corrected for the barotropic tide, SSH, black) with regard to the different dynamics that are separated in terms of frequency bands: subinertial ($\omega < f$, $\text{SSH}_{\text{subinertial}}$, white) for mesoscale and submesoscale dynamics, superinertial frequencies ($\omega > f$, $\text{SSH}_{\text{superinertial}}$, solid red) for internal gravity waves decomposed into the coherent ($\text{SSH}_{\text{coh}}$, blue), incoherent ($\text{SSH}_{\text{inc}}$, yellow) internal tide, and supertidal frequencies ($\omega > 1/10$ h, $\text{SSH}_{\text{supertidal}}$, gray). The altimetry-like SSH corrected for both the barotropic and baroclinic tide and filtered for motions at superinertial frequencies ($\text{SSH}_{\text{corr\_superinertial}}$, dashed red) is also given. The characteristic wavenumber slopes $k^{-2}$ and $k^{-4}$ are represented by the dotted black lines encompassing the mesoscale band (70-250 km), vertical dotted black lines). The transition scale $L_t$ (i.e., where $\text{SSH}_{\text{superinertial}} > \text{SSH}_{\text{subinertial}}$) and the transition scale corrected for $\text{SSH}_{\text{coh}}$ ($L_t^{\text{corr}}$, i.e., where $\text{SSH}_{\text{corr\_superinertial}} > \text{SSH}_{\text{subinertial}}$) for the annually averaged SSH spectra are specified by the red colored numbers.





sect, the incoherent contribution features seasonality and increases in summer, but remains weaker than the coherent signal. Increasing SSH observability of mesoscale and submesoscale motions by correcting for the coherent internal tide signal proves

overall to be more efficient since the incoherent internal tide signal is largely reduced in SSH variance compared to the southern transect (Fig. 9a-b). Specifically, the transition scale is reduced from 211 km to 146 km in summer (Fig. 9a) and from 207 km to 85 km in winter (Fig. 9b). Contributions by motions at supertidal frequencies appear to have larger importance in the northern domain compared to the southern domain. In fact, at scales below 146 km and 85 km for summer and winter respectively, SSH variance is governed by equal contributions from the incoherent internal tide and motions at supertidal frequencies.


Briefly summarized, the dominance of unbalanced motions in the mesoscale to submesoscale band strongly limits SSH observability of geostrophic dynamics around New Caledonia, especially in summer. In other words, SSH observability of mesoscale dynamics is limited to large eddy scales even after a correction for the coherent tide in numerical simulation output. It is to a large part the incoherent internal-tide and non-tidal internal gravity waves at scales $< 100$ km which may eventually constrain

SSH observability of mesoscale and submesoscale dynamics.

### 6.3 SSH wavenumber spectra in along-track direction: a satellite altimetry perspective

The SSH wavenumber spectra in Fig. 9 and the associated conclusions for transition scales are only valid for a one-dimensional transect in tidal beam propagation direction, in which the internal-tide signature is well captured. Here, we mimic SSH wavenumber spectra from satellite altimetry by interpolating SSH from CALEDO60 (similarly to Sect. 6.3) onto realistic

altimetry tracks, which are not aligned with the tidal beam propagation direction (Fig. 10). Further, the SSH wavenumber spectra are averaged for the region south of New Caledonia (Fig. 10a).

South of New Caledonia, and along the tidal beam propagation direction, the coherent tide was found to clearly dominate over subinertial motions or be of comparable importance, regardless of the season (see Fig. 9c-d). However, averaged SSH

wavenumber spectra along given altimetry tracks reveal a different picture. Spectral peaks associated with tidal motions are less pronounced (Fig. 10b-c). $L_t$ is generally decreased, from 175 km to 115 km in summer months and from 165 km to 41 km in winter months. This reduction is linked to the anisotropic nature of the (coherent) internal tide. Along the altimetry track direction, only a fraction of the internal tide energy is captured, causing the SSH wavenumber spectra to emphasize the more isotropic balanced flow regime. Moreover, the incoherent internal tide becomes increasingly important, dominating over the

coherent tide across all spatial scales. This shift reflects the greater isotropy of incoherent SSH. Consequently, in along-track spectra, the dominance of balanced motions and the incoherent internal tide renders corrections for the coherent tide ineffective. This is evident when comparing $L_t$ and $L_t^{\mathrm{corr}}$ in Fig. 10b and c (from 115 km to 107 km in summer months and 41 km to 40 km in winter months).

We conclude that the orientation of altimetry tracks has important implications for the interpretation of transition scales computed from on along-track SSH wavenumber spectra. This effect is particularly pronounced in regions with prominent internal-





tide motion and well-defined propagation beams, such as around New Caledonia. In such cases, transition scales may lead to erroneous estimates of the wavelength at which unbalanced motion becomes dominant over balanced motion, as anisotropic motions like internal tides are not effectively captured in along-track direction. Separating balanced from unbalanced motions is critical for the dynamical interpretation of SWOT SSH, particularly for accessing the mesoscale to submesoscale flow regime and derived quantities such as surface geostrophic velocities. Therefore, caution is advised when using transition scales derived from along-track SSH to assess these dynamics.

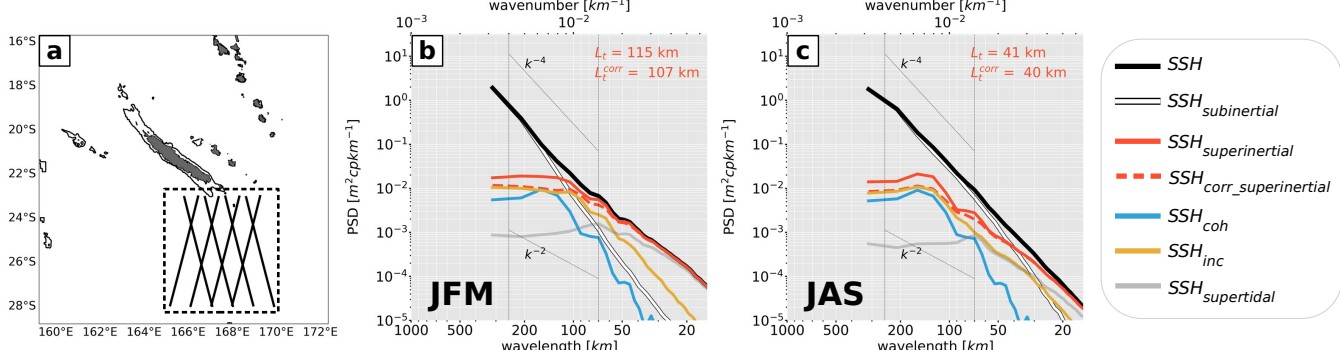

**Figure 10.** Same as in Fig. 9, but for CALEDO60 SSH interpolated onto satellite ground tracks, which reassemble those of SWOT, and averaged over the region south of New Caledonia as shown by the highlighted tracks in (a), separating between (b) summer and (c) winter months.

## 7 Summary and Perspectives

New Caledonia, an archipelago in the southwestern tropical Pacific, is a low-mode semidiurnal internal-tide generation hot spot as revealed by numerical simulation output from a regional model (Bendinger et al., 2023) and in-situ observations (Bendinger et al., 2024). This region is of particular interest for the SWOT altimeter mission since internal tides coexist with the mesoscale to submesoscale circulation. Being subject to potential eddy-internal tide interactions, New Caledonia represents a challenge for SWOT SSH observations. In this companion paper, we investigated temporal variability of the semidiurnal internal-tide, not previously considered in Bendinger et al. (2023). Based on hourly numerical simulation output of a full-model calendar year and a bandpass-filtering technique and harmonic analysis, we decomposed the depth-integrated semidiurnal barotropic-to-baroclinic conversion, energy flux, and dissipation (residual) into their coherent and incoherent parts. These findings are summarized below.




## 7.1 Tidal incoherence in the near-field

In the annual mean, semidiurnal barotropic-to-baroclinic conversion is largely dominated by the coherent tide, which in turn explains a large part of the semidiurnal variability (90-99 %) through the astronomically-forced fortnightly spring-neap tide cycle, i.e., the interaction of M2 and S2. The incoherent tide is negligibly small in the annual mean, suggesting that incoherent contributions cancel out in the long-term average. Though, locally and on shorter time scales, it can explain a notable fraction of semidiurnal variability. Our objective was to identify the underlying mechanisms responsible for conversion variations not

linked to the spring-neap tide cycle.

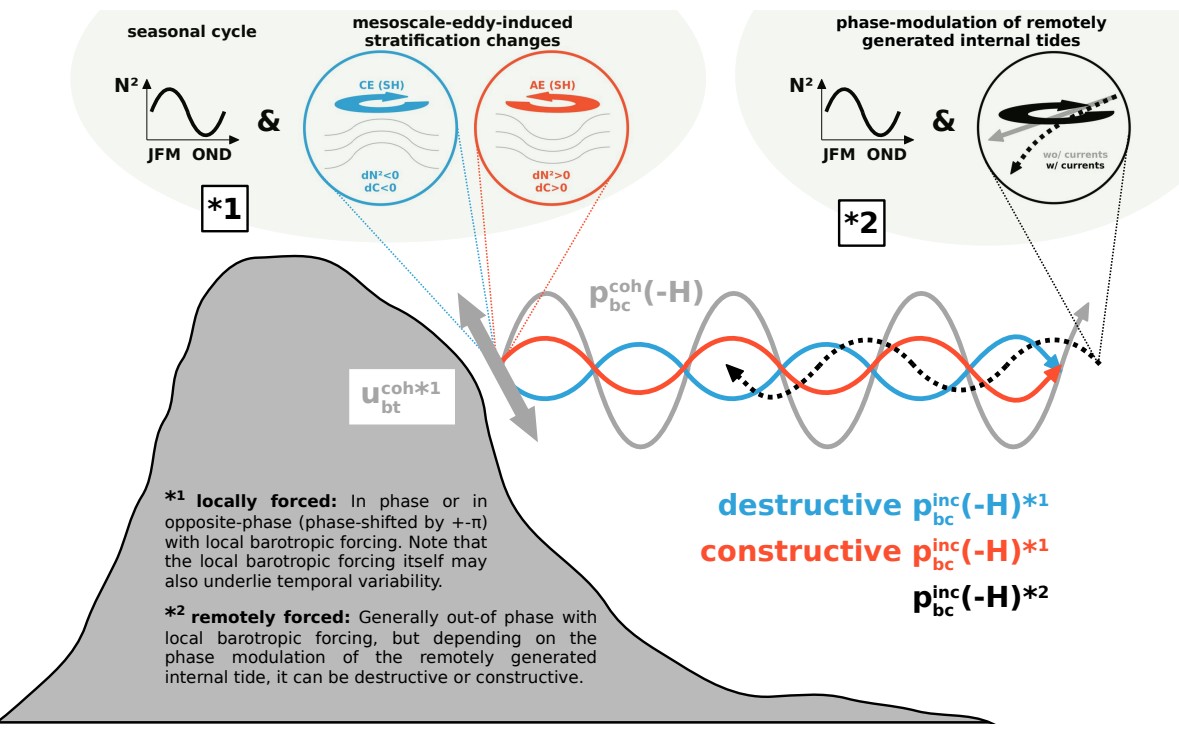

**Figure 11.** Sketch summarizing potential processes that drive conversion variability, focusing on conversion variations due to baroclinic bottom pressure variations $p_{\mathrm{bc}}^{\mathrm{inc}}(-H)$. $u_{\mathrm{bt}}^{\mathrm{coh}}$ is representative of the tidal forcing with the associated baroclinic bottom pressure signal $p_{\mathrm{bc}}^{\mathrm{coh}}(-H)$. Potential mechanisms are separated by local **[*1]** and remote **[*2]** effects. Local effects include stratification changes induced either by seasonal variability or mesoscale eddies. Note that the schematic illustrating mesoscale-eddy induced stratification changes is valid for the Southern Hemisphere (SH). The blue (red) curve represents the incoherent pressure signal $p_{\mathrm{bc}}^{\mathrm{inc}}$ induced by CE (AE), which adds destructively (constructively) to $p_{\mathrm{bc}}^{\mathrm{coh}}$ (gray). Remote effects include the shoaling of remotely generated internal tides (dotted black curve), which are phase-modulated along their propagation path (e.g. tidal beam refraction by mesoscale currents, seasonal stratification changes, etc.). See Sect. 7.1 for a detailed description.





Generally speaking, sources of conversion variations are numerous and are difficult to distinguish due to the unpredictable nature of local and remote effects, summarized in Fig. 11. To well distinguish between these dynamics, we separated the incoherent conversion $C^{\mathrm{inc}}$ into its purely incoherent term ($C^{\mathrm{inc}^*}$) and two cross terms ($C^{\mathrm{cross1}}$, $C^{\mathrm{cross2}}$) (see Equation 6).

Our analysis suggests that the work done by the barotropic tide on baroclinic bottom pressure variations dominates incoherent conversion (71-90 %), expressed by $C^{\mathrm{cross1}}$. The majority occurrence of baroclinic bottom pressure amplitude variations (over phase variations) imply local effects to govern over remote effects. Locally and on monthly to intraseasonal time scales, mesoscale-eddy-induced stratification changes through upward (CE) and downward (AE) pumping of isopycnal surfaces can induce negative and positive conversion anomalies by more than 20 %, respectively. This is supported by positive correlations

of baroclinic bottom pressure amplitude variations with bottom stratification and mesoscale SLA. Seasonal variability seems to play a minor role or is superimposed on the dominant mesoscale variability. The importance of conversion variations through changes in the barotropic tide forcing remains to be investigated, but we showed that they can explain an important fraction by up to 23 % in the incoherent conversion term.

## 7.2 Tidal incoherence in the far-field

Semidiurnal tidal energy propagating towards the open ocean follows at first order the spring-neap tide cycle, closely coupled to semidiurnal conversion variability. However, it features elevated levels of tidal incoherence, which increases with increasing distance to the generation sites. Close to the generation sites, tidal incoherence explains up to 20 % of the semidiurnal variability. In the far-field, it can account for up to 90 % (in 500-1000 km distance to the generation site). Tidal incoherence of the semidiurnal energy flux is here associated with the refraction of the tidal beams in propagation direction. This is in agreement

with a simplified ray tracing, which tracks the horizontal propagation of inertia-gravity modes at semidiurnal frequency with varying (depth-independent) background currents. The refraction is linked with varying group and phase speeds induced by the background currents, which in turn cause the tidal rays to change orientation. Tidal incoherence is generally of higher importance south of New Caledonia corresponding with enhanced mesoscale activity, leading to increased dispersion and phase variability there. North of New Caledonia, the theoretical rays closely align with the theoretical propagation direction for a

semidiurnal ray in the absence of background currents. This, in turn, is in agreement with reduced mesoscale activity.

## 7.3 Incoherent-tide SSH limits SSH observability of mesoscale to submesoscale motions

The dynamical interpretation of SSH in regions where balanced and unbalanced motions feature similar SSH variance at comparable wavelengths is challenging. By revisiting Part 1, we computed SSH wavenumber spectra in tidal energy propagation

direction and extended the analysis by investigating the relative importance of the incoherent tide in SSH variance. A correction for the coherent SSH is only partly effective to access scales toward smaller wavelengths due to superinertial and subinertial motions being out of phase, i.e., superinertial motions are enhanced in summer while subinertial motions are considerably reduced and vice versa in winter. Ultimately, it is the incoherent tide, which limits SSH observability of balanced and unbalanced





motions to scales above 150 km in summer and above 80 km in winter.


SSH wavenumber spectra computed along altimetry tracks lead to several conclusions. The altimetry tracks are not oriented in tidal energy propagation direction and, therefore, the spectra capture only part of the internal-tide energy. The incoherent tide dominates over the coherent tide across all wavelengths, reflecting its more isotropic nature compared to the coherent tide. Moreover, transition scales derived along altimetry tracks are generally reduced compared to those determined in tidal energy

propagation direction since the SSH wavenumber spectra emphasize the more isotropic balanced flow regime. Relying on these transition scales in regions where balanced and unbalanced motions coexist may result in a distorted view of the governing dynamics as anisotropic processes such as internal tides are not properly sampled. Yet, knowing at which wavelengths balanced and unbalanced motions dominate is particularly crucial for SWOT in order to disentangle the different flow regimes in SSH measurements.

## 7.4 Perspectives of this work

This study provides several routes for future work to increase our understanding of the internal-tide life cycle. An open question that arises from our analysis concerns the relationship between conversion variability and the impact on outward energy propagation and local energy dissipation. Specifically, is energy flux divergence directly correlated with conversion, or are their variations partially decoupled? Similarly, what can be inferred about dissipation - does higher conversion consistently enhance

dissipation, and do negative anomalies reduce it (Falahat et al., 2014a)? Monthly anomalies of $C^{\mathrm{D2}}$, $\boldsymbol{\nabla}_{\mathrm{h}} \cdot \mathbf{F}^{\mathrm{D2}}$, and $D^{\mathrm{D2}}$ relative to the annual mean are shown in Fig 12. Energy flux divergence anomalies generally follow those of conversion, with positive

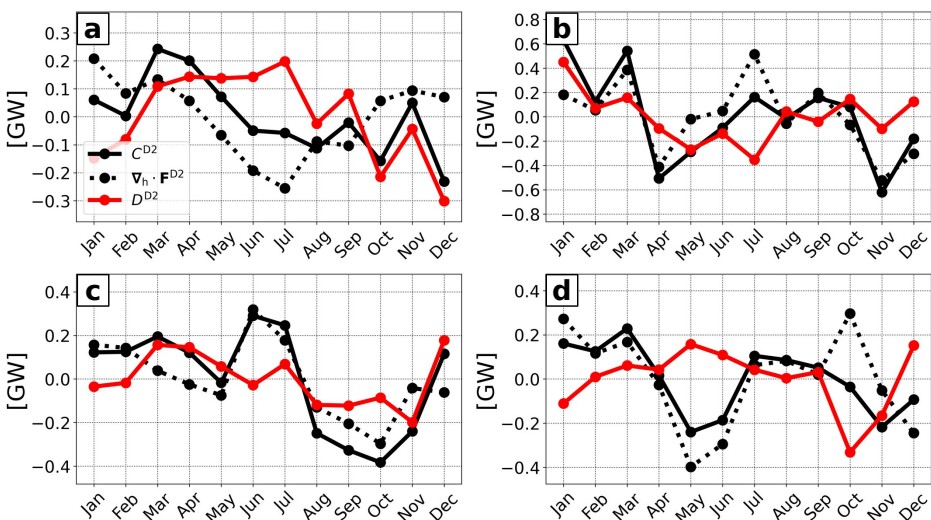

**Figure 12.** Monthly time series of semidiurnal barotropic-to-baroclinic conversion $C^{\mathrm{D2}}$ and energy flux divergence $\boldsymbol{\nabla}_{\mathrm{h}} \cdot \mathbf{F}_{\mathrm{bc}}^{\mathrm{D2}}$, and $D_{\mathrm{bc}}^{\mathrm{D2}}$ anomalies relative to the annual mean, integrated over (a) North, (b) South), (c) Norfolk Ridge, (d) Loyalty Ridge.





conversion anomalies typically resulting in increased energy flux divergence, and vice versa. In contrast, dissipation anomalies are less variable. Nevertheless, periods exist when energy flux divergence anomalies are either more or less pronounced than conversion anomalies, leading to reduced or enhanced dissipation, respectively. The processes governing whether excess or deficit tidal energy conversion is balanced primarily by outward energy propagation or by local dissipation (or other terms in Equation 1 remain to be fully understood.

This highlights the need for further investigation into the mechanisms controlling energy partitioning between these pathways. Those findings could have important implications for parameterizations of internal-tide dynamics such as tidal mixing and dissipation for climate and ocean general circulation models, which do not resolve tidal processes. Current parameterizations consider geographically varying tidal mixing (Vic et al., 2019; de Lavergne et al., 2019, 2020). However, temporal variations induced by the spring-neap cycle, mesoscale variability, and seasonal changes are not taken into account.

Further insight is expected from an extensive in-situ experiment (SWOTALIS, Cravatte et al., 2024) that was carried out in March-May 2023. It was inter-alia dedicated to the deployment of full-depth oceanographic moorings and located in the hot spots of internal-tide generation and dissipation south of New Caledonia. Successfully recovered in November 2023, these moorings provide a unique dataset to better understand the internal-tide life cycle, while assessing our numerical model output. Furthermore, this region is located beneath the two swaths of SWOT's fast-sampling phase (1-day repeat orbit). The moorings and the numerical simulation output will play an essential role in the dynamical interpretation of SWOT SSH by allocating the different dynamics such as balanced and unbalanced motions. Emphasis will be given to the SSH signature of the incoherent internal tide, which represents a major challenge for SWOT SSH observability.

**Appendix A**

The hypothesis of conversion variations driven by mesoscale-eddy-induced stratification changes (see Sect. 4.1) is further supported below. We illustrate how mesoscale eddies, AE and CE, may affect the bottom stratification at the internal-tide generation site. Two examples, shown in Fig. A1, illustrate the presence of either an AE (Fig. A1a) or a CE (Fig. A1b) above Norfolk Ridge (3). These features are represented by positive and negative monthly-averaged sea level anomalies (SLA) for July and August, respectively. The zonal sections of the corresponding isopycnals indicate mesoscale-eddy-induced downward and upward pumping. The resulting stratification changes are depicted in zonally averaged stratification profiles (gray: annual mean; red: July; blue: August). In July, the AE causes downward pumping of isopycnals, leading to positive stratification anomalies near the seafloor (Fig. A1a). Conversely, in August, the CE induces upward pumping of isopycnals, which results in negative stratification anomalies near the seafloor (Fig. A1b). It is noteworthy that the stratification anomalies just above the seafloor (at approximately 800 m depth) are significantly smaller than those in the water column above. However, they consistently align with the overall sign of the stratification anomaly.



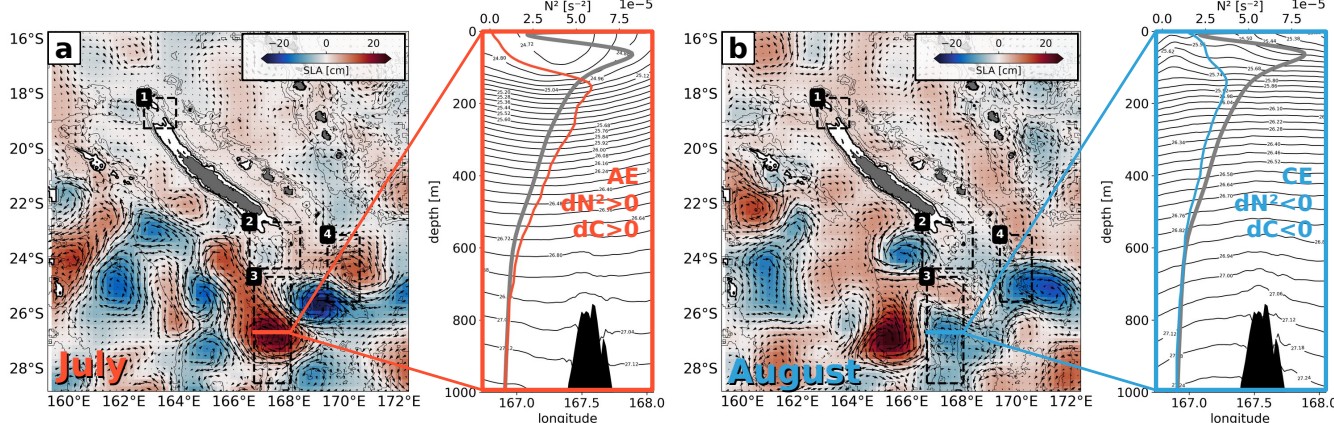

**Figure A1.** Monthly averaged mesoscale SLA for (a) July and (b) August showing qualitatively the impact of mesoscale variability on stratification and conversion through downward and upward pumping induced by AE and CE, respectively. The pumping is illustrated by the isopycnals along a zonal section trough the AE and CE. The zonal mean stratification profile for the given month (red: July; blue: August) as well as the annual mean (gray) is also shown.

The monthly time series of the area-integrated conversion anomaly including domain-averaged baroclinic bottom pressure amplitude and phase variations as well bottom stratification, mesoscale SLA and EKE for each subdomain are shown in Fig. A2. In the South (2) and Norfolk Ridge (3) domains, the above hypothesis is strengthened by the positive correlation coefficients (Fig. A2b and A2c). Even though we primarily link conversion variations to mesoscale variability, the monthly time series partly suggest seasonal variations of conversion. It tends to be enhanced in summer months and reduced in winter months corresponding with seasonally varying stratification. Whatsoever, mesoscale variability clearly enhances, suppresses or even reverses the seasonally-driven anomalies. A clear distinction between seasonally- and mesoscale-driven stratification changes is needed in future work to allocate the exact contribution to conversion variations.

*Code and data availability.* The tidal analysis was performed using the COMODO-SIROCCO tools which are developed and maintained by the SIROCCO national service (CNRS/INSU). SIROCCO is funded by INSU and Observatoire Midi-Pyrénées/Université Paul Sabatier and receives project support from CNES, SHOM, IFREMER and ANR (https://sirocco.obs-mip.fr/other-tools/prepost-processing/comodo-tools/, last access: 25 August 2023). The numerical model configuration (CALEDO60) used in this study is introduced and described in detail in Bendinger et al. (2023). The data to reproduce the figures can be found in Bendinger (2025a) with the associated scripts in Bendinger (2025b). The ray-tracing algorithm is described in full detail in Sect. 3b in Rainville and Pinkel (2006).



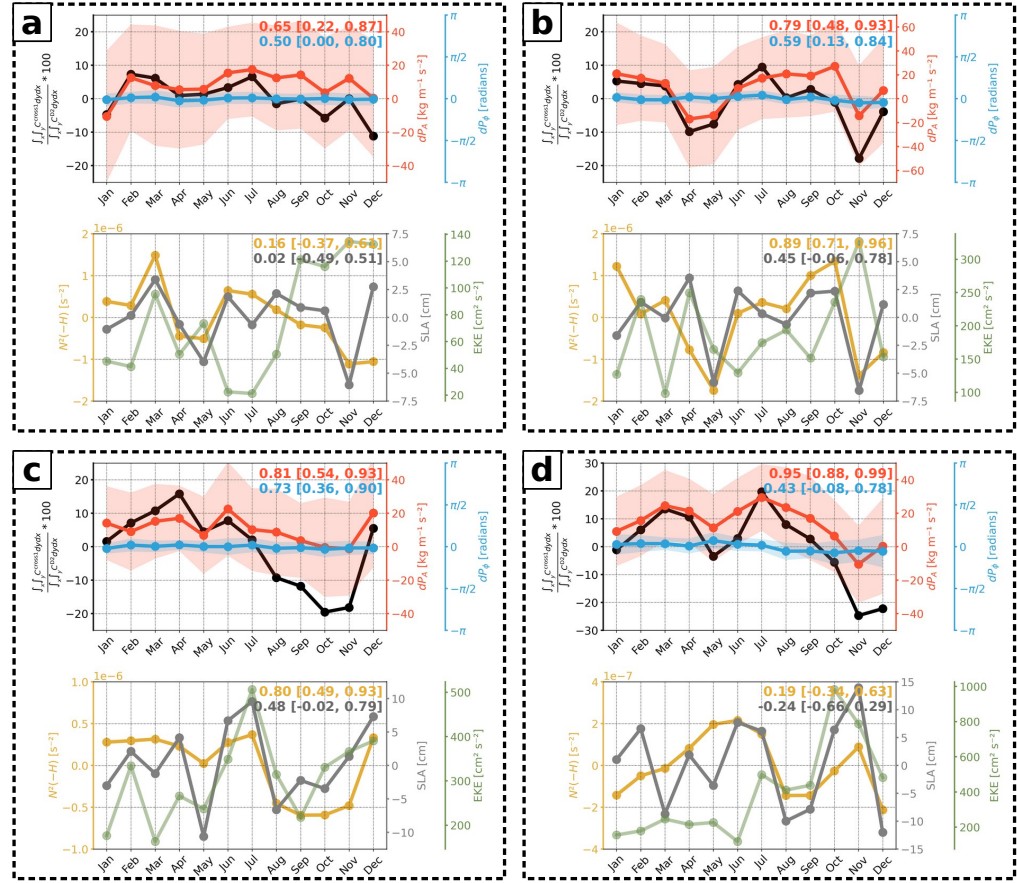

**Figure A2.** Monthly time series of the area-integrated conversion anomaly (black) induced by $C^{\mathrm{cross1}}$ expressed by the ratio of $C^{\mathrm{inc}}$ and $C^{\mathrm{D2}}$, domain-averaged baroclinic bottom pressure amplitude ($dP_A$, red) and phase ($dP_\phi$, blue) difference between the semidiurnal and coherent tide (including standard deviation), domain-averaged bottom stratification ($N^2(-H)$, yellow), mesoscale SLA (gray) and EKE (green) for (a) North (1), (b) South (2), (c) Norfolk Ridge (3), Loyalty Ridge (4). The correlation coefficients of $C^{\mathrm{cross1}}$ with $dP_A$, $dP_\phi$ and of $dP_A$ with $N^2(-H)$, mesoscale SLA including their 90 % confidence intervals are also given.

*Author contributions.* AB performed the analysis and drafted the manuscript under the supervision of LG and SC. CV and FL were deeply involved in the discussion and interpretation of scientific results. All co-authors reviewed the manuscript and contributed to the writing and final editing.

*Competing interests.* The authors declare that they have no conflict of interest.



*Acknowledgements.* This work has been supported by the Université Toulouse III - Paul-Sabatier (grant from the Ministère de l'Enseignement supérieur de la Recherche et de l'Innovation, MESRI) carried out within the PhD program of Arne Bendinger at the Faculty of Science and Engineering and the Doctoral School of Geosciences, Astrophysics, Space and Environmental Sciences (SDU2E). Sophie Cravatte and Lionel Gourdeau are funded by the Institut de Recherche pour le Développement (IRD); Clément Vic was funded by the Institut français de recherche pour l'exploitation de la mer (IFREMER); Florent Lyard was funded by the Centre National de la Recherche Scientifique (CNRS); RC was funded by CLS. This study has been partially supported through the grant EUR TESS N°ANR-18-EURE-0018 in the framework of the Programme des Investissements d'Avenir. This work is a contribution to the joint CNES-NASA project *SWOT in the Tropics* and is supported by the French TOSCA (la Terre, l'Océan, les Surfaces Continentales, l'Atmosphère) program and the French national program LEFE (Les Enveloppes Fluides et l'Environnement).



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
