# Peer review of "Regional modeling of internal-tide dynamics around New Caledonia. Part 2: Tidal incoherence and implications for sea surface height observability"

_EGUsphere, 2025_

## Referee Comment (RC1)

**Review of "Regional modeling of internal-tide dynamics around New Caledonia. Part 2: Tidal incoherence and implications for sea surface height observability"**

***by Bendinger et al, submitted for publication in Ocean Science***

This paper investigates the dynamics and incoherence of the internal tide around New Caledonia using a year-long, high-resolution, realistic numerical simulation at 1/60°.

The present paper is a follow-up to a previously published paper (Bendinger et al. 2023, referred to as "Part 1") where the authors focused on the coherent internal tide in the same region, mostly based on the same numerical simulation. Here, the authors address three main aspects related to the incoherent tide: 1) the role and nature of the variability of the tidal conversion term ("local incoherence"), 2) the impact of mesoscale variability on the loss of coherence of the internal tide during its propagation ("far-field incoherence"), and 3) the implications of the non-isotropy and incoherence of the internal tide for the definition of a transition length scale between balanced and unbalanced flow. The latter point is related to the use of altimeter data and the issue of disentangling balanced from unbalanced flow, and is timely in the context of the recently launched SWOT mission.

The manuscript is well written, if somewhat long. The methodology is sound and generally well described. The manuscript addresses different issues (two related to IT dynamics and incoherence, one focusing on the implications for SSH imprints), and I think that in some places - which I flag below - the authors could shorten the text. Nevertheless, it is an interesting contribution to the field of physical and observational oceanography, and **I support its publication in Ocean Science**. I do have a few comments, which I call "minor" because they do not compromise the coherence and nature of the results reported.

**Comments & questions**

**Decomposition of the conversion term (eq. 6 and subsequent paragraph)**

By construction, when the coherent tide is extracted by least-squares regression, the time average (over the same period as the coherent extraction) of a bilinear product between a coherent field and an incoherent field is expected to be close to zero (minimal in a least-squares sense if the relative magnitude of the different coherent frequencies is similar for the different fields) – see Savage et al. (2020), citing Wunsch (2006).

**Section 4.1.**

- While the authors only mention the influence of varying stratification as a cause of local changes in the baroclinic bottom pressure and, incidentally, the barotropic conversion term, it should be noted that the direct influence of background currents on generation has also been demonstrated in the literature (Dunphy & Lamb 2018, Shakespeare 2020, Dossman et al. 2020).
- Please specify whether $dP_A$ and $dP_\phi$ are the amplitude and phase of the incoherent baroclinic bottom pressure $P - P_{\text{coh}}$, or whether it is the difference of the amplitude and phase of the total (coherent + incoherent) and coherent baroclinic bottom pressure. Also, is it possible that the phase term cancels out when averaging over the subdomains?

**Section 5 and 2.3**

- I am not convinced of the relevance of the results described in this section. I encourage the authors to clarify how this analysis adds to our understanding of internal tidal dynamics and incoherence. What is the added value of the ray tracing approach? No quantitative information is derived from it, the only interpretation it is that the ray paths are more scattered in the south compared to the north, which is sort of expected by nature, given the differences of mesoscale activity. Furthermore, it does not provide any significant additional information compared to the results previously presented in Bendinger et al. (2024). Also, the authors mention that the role of variations in phase velocity (related to stratification) remains unknown, but I think it could be incorporated into their ray tracing approach by using the model stratification instead of a climatology (see e.g. Zaron & Egbert 2014). I agree that several papers have found that background currents have a dominant effect on IT refraction, but not that the latter is negligible - to my understanding. Unless this section is enriched or the relevance of the results emphasised, I would suggest that this section be shortened.
- Please describe the ray tracing experiments in more detail: do you start a ray from each starting point every day to get the different ray paths? How many paths are computed?
- Section 2.3: since this is essentially copied and pasted from Bendinger et al (2024), I suggest shortening this section – possibly keeping the detailed description as an appendix.

**Figure 11 in Summary**

I am rather sceptical that this figure helps to provide a synthetic view of the results. I think the text is sufficient (and even clearer).

**Others, general or specifics**

- I do not think that internal tide needs to be hyphenated in the title
- The introduction touches on various aspects of internal incoherence, but does not give much detail on what is known in the region. I understand that the regions have not received much attention until recently (in terms of IT dynamics and incoherence), but it could be seen as a synthesis of what is known. For example, I noticed that New Caledonia can be identified in the incoherent variance fraction estimates of Zaron et al. (2017) or Nelson et al. (2019).

- Eq. 1: Please briefly explain where this equation comes from (and/or add relevant references).
- L. 208–210: There is an incoherence here: depth-independent currents correspond only to a barotropic structure, as a mode-1 baroclinic current as a vanishing vertical integral (or average). My understanding is that using depth-independent currents for the ray-tracing approach is rather a simplification, because including the effect of the vertically sheared background current is complicated – unless one is willing to implement a much more complicated approach, as in Duda et al. 2018 or Guo et al (2023)
- On l.304, the authors claim that local effects associated with local stratification changes are associated with a modulation of the amplitude of the baroclinic bottom pressure, while remote effects are associated with pressure phase variations. Is there any evidence or previous work to support this claim? While I suppose the former can be anticipated from linear theory (e.g. following St Laurent & Garrett 2002), I do not see a straightforward explanation for the impact of remote waves being associated only with phase variations.
- l.325: Again, how are variations in near-bottom stratification and bottom pressure related? Is this straightforward from the above equation for $p(z)$?
- Figure 5 caption: missing parenthesis at end of first sentence.
- l.409: the delay is in hours, not days. Also, the delays obtained are shorter in the south than in the north, which is somewhat contradictory with the fact that IT is more incoherent there.

**References (not mentioned in the manuscript)**

- [Dossmann et al 2018]: Dossmann, Y., Shakespeare, C., Stewart, K. & Hogg, A. McC. Asymmetric Internal Tide Generation in the Presence of a Steady Flow. Journal of Geophysical Research: Oceans 125, e2020JC016503 (2020).
- [Dunphy & Lamb 2018]: Lamb, K. G. & Dunphy, M. Internal wave generation by tidal flow over a two-dimensional ridge: energy flux asymmetries induced by a steady surface trapped current. Journal of Fluid Mechanics 836, 192–221 (2018).
- [Nelson et al 2019]: Nelson, A. D. et al. Toward Realistic Nonstationarity of Semidiurnal Baroclinic Tides in a Hydrodynamic Model. J. Geophys. Res. Oceans 124, 6632–6642 (2019).
- [St Laurent & Garrett 2002]: St. Laurent, L. & Garrett, C. The Role of Internal Tides in Mixing the Deep Ocean. Journal of Physical Oceanography 32, 2882–2899 (2002).
- [Savage et al 2020]: Savage, A. C., Waterhouse, A. F. & Kelly, S. M. Internal tide nonstationarity and wave-mesoscale interactions in the Tasman Sea. J. Phys. Oceanogr. 1–52 (2020) doi:10.1175/JPO-D-19-0283.1.
- [Shakespeare 2020]: Shakespeare, C. J. Interdependence of internal tide and lee wave generation at abyssal hills: global calculations. J. Phys. Oceanogr. (2020) doi:10.1175/JPO-D-19-0179.1.
- [Wunsch 2006]: Wunsch, C. Discrete Inverse and State Estimation Problems: With Geophysical Fluid Applications. (Cambridge University Press, Cambridge, 2006). doi:10.1017/CBO9780511535949.
- [Zaron & Egbert 2014]: Zaron, E. D. & Egbert, G. D. Time-Variable Refraction of the Internal Tide at the Hawaiian Ridge. Journal of Physical Oceanography 44, 538–557 (2014).

---

## Referee Comment (RC2)

The study describes the temporal variations of internal tide around New Caledonia from an energy perspective and serves as a companion to Bendinger et al. (2023). The authors conclude that mesoscale eddies contribute to the variations of internal tide generation, as shown through the decomposition of the incoherent term $C^{inc}$, and propagation, analyzed using a ray-tracing model. Specially, at the generation site (the near field), variations in conversion are linked to changes in baroclinic bottom pressure, which correlate positively with mesoscale-induced stratification changes. After propagation (the far field), regions with strong eddy activity exhibit enhanced incoherent tides and refraction in the propagation direction. The study also explores implications for SSH observability, highlighting that incoherent tides and the orientation of altimetry tracks can impact the separation of balanced and unbalanced motions.

Overall, the study is well-executed, providing solid evidence for the proposed mechanisms through quantified analysis. The discussion on the impact of altimetry track orientation on SSH wavenumber spectra is novel and relevant to the SWOT mission. However, the manuscript is somewhat lengthy, and certain sections could be condensed for clarity. Additionally, I have several questions and comments that would like to be addressed before publication.

**Introduction**

- The recap for Part 1 (l.34-54) can be shorten by focusing on those related to incoherence.
- The energy dissipation is introduced in l.71-81, and understanding its temporal variation is stated as one of the objectives in l.106. However, the analysis of energy dissipation is only addressed at the annual mean timescale and described in a single paragraph (l.241-251). The emphasis

on the dissipation term in the introduction does not align with the following analysis focus.

**Section 2**

- Equation 1 can be omitted to maintain relevance and save space.
- Equation 7, why is the energy flux intergraded from 0 instead of $\eta$?
- Equation 11, the right parathesis and "dt" for $D_{bc}^{coh}$ term are missing.
- At the end of Section 2.2, the authors state that the $D_{bc}^{inc}$ represents the overestimated portion of $D_{bc}^{coh}$. A similar statement appears in Section 3 (l. 246-247). However, I think the $D_{bc}^{inc}$ includes both the overestimated coherent portion AND the actual dissipation from the incoherent tide. The presence of real incoherent dissipation can be verified by the net incoherent dissipation at North (1). Nevertheless, the conclusion regarding the overestimation remains unchanged.
- Section 2.3 is similar to Bendinger et al (2024) and can be shorten.

**Section 4**

- It is unusual to have only one subsection. I suggest reorganizing the structure. The same applies to Section 5.
- The explained variability is expressed as (%) in text (e.g. l.259-260) while as decimal (0-1) in Table 2. Please ensure consistency.
- L.315: "The negative conversion/bottom pressure amplitude anomalies in Fig. 5b". Should likely refer to "Fig. 5a"?
- L.327: "In phase with the local tidal forcing, $p_{bc}^{inc}(-H)$ induced ...". If $p_{bc}^{inc}(-H)$ is "in-phase", why it is incoherent?

**Figure 3**

- The bathymetric labels are too small to be clearly visible.

**Figure 5**

- 5(a): What is the physical meaning of the negative values for the ratios of $C^{cross1}/C^{D2}$ on the y axis?

**Table 3**

- The unit for "delay" should be [days].

**Section 5**

- The "group arrival time" is not clearly defined in Section 5.1. Based on the caption in Table 3, which states "equivalent to 500 km", I assume the "group arrival time" refers to the time taken to propagating 500 km. If so, for South (2) domain, both mode-1 and mode-2 tide propagate faster with the mesoscale currents, which is consistent with the negative "delay" time. If my understanding is correct, the statement in l.409-410, "Mode 2 is substantially *more delayed* than mode 1", is incorrect for the South (2) domain, as both mode-1 and mode-2 *arrive earlier*, with negative delayed time.
- I also speculate on the "delay" for South (2) domain. The standard deviation is large enough to cause the "delay" time to switch signs.

**Section 6**

- The paragraphs before 6.1 (l.421-445) serve as a recap in Part 1 and an introduction, so they can be condensed. The same applies to l.459-465 in Section 6.2.

Below are some minor comments, which I leave to the authors' discretion to consider.

**Vague pronoun reference**: please explicitly state the references for clarity.

- L.10: "it"
- L.77: "this can have..."
- L.141: "their Fig. 13 a-d"
- L.259: "it explains"

**Overuse of "i.e.":** here are some suggested replacements.

- L.141: "... Caledonia, **including** the location ..."
- L.163: "... incoherent parts **for** u and the pressure perturbation p"
- L.255: "... spring-neap cycle, **driven by** the interaction of M2 ..."

**Formatting and grammar corrections:**

L.5: "...from coherence, in...", add space after the comma.

L.8 & l.255: The phrase "astronomically forced fortnightly modulated spring-neap cycle" is wordy and grammatically incorrect, which reduces readability. Consider a revision.

L.9: use an "em dash" rather than a hyphen. The same applies to others in the manuscript.

L.29: "semi-analytical theory"

L.45: "representative **to** the coherent"

L.57: I think "near-field" and "far-field" that have a hyphen in between, are commonly used as adjectives, rather than nouns.

L.64-65: suggested revision "The mechanisms governing the temporal variability of internal tide vary geographically, and cannot be generalized as the importance …"

L.67-69: suggested revision "New Caledonia is a particularly challenging region as it is a hot spot of internal tide generation and a region of strong mesoscale variability, making it potentially …"

L.94: "estimate the length scale **at which** unbalanced motions …"

L.186: "taken from the harmonic analysis and vertical mode …"

L. 215: missing the right parathesis after "(see Fig.1)"

L. 220: "mimic"

L.239: "distance **from** the generation"

L.247: "This **accounts for** 10%, 9% …"

L.276 (Figure 3 caption): The second sentence lacks a verb.

L.280: "Similarly to the **analysis** above, we show **in** the South (2) domain, the contribution of different terms that make up $C^{inc}$ …"

L.282-283: "**While** the three terms feature similar amplitudes, **their** spatial patterns differ."

L.283: "Based on the area-integrated explained … "

L.317 & 319: "compute 5-day mean" and "period of 180 days"

L.324: "conversion and mesoscale variability …", which conversion?

L.335: "pressure amplitude variations are very pronounced**, suggesting** the influence of the local effects"

L.379: "closely correlated to **that** of semidiurnal"

L.612: "concerns the impact of conversion variability on outward energy propagation and local energy dissipation"? The logic between these three terms in Equation 2 is unclear to me.

---

## Author Comment (AC1)

**Regional modeling of internal-tide dynamics. Part 2: Tidal incoherence and implications for sea surface height observability**

**Reviewer #1**

This paper investigates the dynamics and incoherence of the internal tide around New Caledonia using a year-long, high-resolution, realistic numerical simulation at 1/60°. The present paper is a follow-up to a previously published paper (Bendinger et al. 2023, referred to as "Part 1") where the authors focused on the coherent internal tide in the same region, mostly based on the same numerical simulation. Here, the authors address three main aspects related to the incoherent tide: 1) the role and nature of the variability of the tidal conversion term ("local incoherence"), 2) the impact of mesoscale variability on the loss of coherence of the internal tide during its propagation ("far-field incoherence"), and 3) the implications of the non-isotropy and incoherence of the internal tide for the definition of a transition length scale between balanced and unbalanced flow. The latter point is related to the use of altimeter data and the issue of disentangling balanced from unbalanced flow, and is timely in the context of the recently launched SWOT mission.

The manuscript is well written, if somewhat long. The methodology is sound and generally well described. The manuscript addresses different issues (two related to IT dynamics and incoherence, one focusing on the implications for SSH imprints), and I think that in some places - which I flag below - the authors could shorten the text. Nevertheless, it is an interesting contribution to the field of physical and observational oceanography, and I support its publication in Ocean Science. I do have a few comments, which I call "minor" because they do not compromise the coherence and nature of the results reported.

First of all, we would like to thank the reviewer for taking the time to go through our manuscript, for the thorough comments and recommendations which helped improve clarity while pointing out the key findings of our study. In the following, we will address the reviewer's comment and state where changes have been made in the manuscript.

**Comments & questions**

**Decomposition of the conversion term (eq. 6 and subsequent paragraph)**

By construction, when the coherent tide is extracted by least-squares regression, the time average (over the same period as the coherent extraction) of a bilinear product between a coherent field and an incoherent field is expected to be close to zero (minimal in a least-squares sense if the relative magnitude of the different coherent frequencies is similar for the different fields) – see Savage et al.

(2020), citing Wunsch (2006).

We appreciate the reviewer's comment. We modified the paragraph for the sake of clarity: "This classification follows from the orthogonality condition inherent in the least-squares framework: the residual (i.e., the incoherent component) is uncorrelated with the fitted coherent signal and lacks a consistent phase relationship with the tidal forcing. In other words, there is no preferred phasing between coherent and incoherent motions. As a consequence, bilinear products between coherent and incoherent fields tend to average out or remain negligibly small over the regression period (Nash et al., 2012; Buijsman et al., 2017; Savage et al., 2020). However, as we show in the analysis below, these cross terms can account for a substantial fraction of the variability on shorter time scales. We place particular emphasis on assessing their relative importance to better understand temporal variations in the conversion term and the physical mechanisms that drive them." See Lines 166-173

**Section 4.1**

While the authors only mention the influence of varying stratification as a cause of local changes in the baroclinic bottom pressure and, incidentally, the barotropic conversion term, it should be noted that the direct influence of background currents on generation has also been demonstrated in the literature (Dunphy and Lamb 2018, Shakespeare 2020, Dossman et al. 2020).

We thank the reviewer for these references. They are relevant for the study. We included the following paragraph: "Other local processes include the direct influence of background currents, which induce asymmetries in internal-tide generation being enhanced on the upstream side of bathymetric obstacles (Lamb and Dunphy, 2018; Shakespeare, 2020; Dossmann et al., 2020)." See Lines 355-357

Please specify whether $dP_A$ and $dP_\phi$ are the amplitude and phase of the incoherent baroclinic bottom pressure $P - P^{\mathrm{coh}}$, or whether it is the difference of the amplitude and phase of the total (coherent + incoherent) and coherent baroclinic bottom pressure. Also, is it possible that the phase term cancels out when averaging over the subdomains?

$dP_A$ and $dP_\phi$ is the amplitude and phase difference between the semidiurnal (coherent + incoherent) and coherent baroclinic bottom pressure. This was mentioned in the caption of Figure 6. But the reviewer is right that this should be clarified earlier. We added the following in Section 4.1: "Here, $dP_A$ and $dP_\phi$ are representative of the amplitude and phase difference between the semidiurnal and coherent tide, determined by complex demodulation of $p_{\mathrm{bc}}^{\mathrm{D2}}(-H)$ and $p_{\mathrm{bc}}^{\mathrm{coh}}(-H)$." See Lines 291-292

Averaging over the subdomains is an interesting aspect since the considered processes can be very local. Both amplitude and phase terms can cancel out when averaging over the domains. For this reason, we decided to show the standard deviation for $dP_A$ and $dP_\phi$ in Fig. 5a. Overall, variations of phase are much reduced compared to variations in amplitude. This also has consequences for the conversion term. For Pines Ridge, the conversion anomaly induced by baroclinic bottom pressure variations is at most 10 %. This ratio can be much higher locally. This is also evident from Fig. 4h showing that those conversion anomalies largely vary in space.

**Section 5 and 2.3**

I am not convinced of the relevance of the results described in this section. I encourage the authors to clarify how this analysis adds to our understanding of internal tidal dynamics and incoherence.

What is the added value of the ray tracing approach? No quantitative information is derived from it, the only interpretation it is that the ray paths are more scattered in the south compared to the north, which is sort of expected by nature, given the differences of mesoscale activity. Furthermore, it does not provide any significant additional information compared to the results previously presented in Bendinger et al. (2024). Also, the authors mention that the role of variations in phase velocity (related to stratification) remains unknown, but I think it could be incorporated into their ray tracing approach by using the model stratification instead of a climatology (see e.g. Zaron and Egbert 2014). I agree that several papers have found that background currents have a dominant effect on IT refraction, but not that the latter is negligible - to my understanding. Unless this section is enriched or the relevance of the results emphasised, I would suggest that this section be shortened.

We thank the reviewer for this suggestion. First of all, we like to point out that even though the ray tracing methodology is similar to Bendinger et al. (2024), it does not follow the same objective. In Bendinger et al. (2024), the ray tracing served a case study to illustrate that the differences in internal-tide amplitude/phase in glider observations and a virtual glider from numerical modeling output are linked to the refraction of the tidal beam through mesoscale currents. In the presented manuscript, the ray tracing was primarily used to quantify tidal beam dispersion in propagation direction to highlight the contrast in tidal incoherence for the regions north and south of New Caledonia. For the sake of statistical robustness, this was done for daily-averaged horizontal velocity fields of a full-calendar year.

However, we agree with the reviewer that the analysis could be enriched without significantly lengthening the section. To do so, we made the following changes in the methodology:

1. We replaced the climatological stratification by the model stratification alongside the model bathymetry.

2. The reference scenario is now based on the annually averaged stratification and depth-averaged currents.

3. Three experiments are conducted: I. Varying stratification and varying currents, II. Varying stratification and annually averaged currents, III. Annually averaged stratification and varying currents.

4. Consistent with the remaining analysis (see Figure 5, 6, 12, A1, A2), we decided to apply the ray tracing experiment from 3. on the monthly-averaged fields of stratification and currents. Previously, we applied the ray tracing on daily-averaged currents. By doing so, we focus exclusively on mesoscale-induced effects as we did for the internal-tide generation.

5. Additionally, we plotted the ray as a function of the phase offset relative to the reference wave.

Based on the enriched analysis, Section 5.1 has been adapted. The main result remains unchanged, i.e., tidal beam dispersion in propagation direction is increased south of New Caledonia, where mesoscale-eddy activity is higher. The current effect is dominant, however, as pointed out by the reviewer, stratification effects are not negligible. To quantify this, we decided to compute the cumulative refraction which is experienced by a ray in propagation direction for the three different scenarios. To do so, we make use of the group velocity vector to compute the angle representative of the ray's changing orientation (see Table 3, and Lines 390-393). Note that we integrate absolute

values of angular deviation to avoid compensating effects along the ray path.

Please describe the ray tracing experiments in more detail: do you start a ray from each starting point every day to get the different ray paths? How many paths are computed?

Initially, we started a ray for each day (for daily currents) for the two initialization regions north and south of New Caledonia. For each initialization region, in total 365 rays were computed. In the revised manuscript, we rely on monthly-averaged stratification and currents as stated in the above comment. Therefore, we obtain only 12 rays per region, but the main message remains unchanged. A more detailed description of the ray experiments has been added to Section 2.3.

Section 2.3: since this is essentially copied and pasted from Bendinger et al (2024), I suggest shortening this section – possibly keeping the detailed description as an appendix.

Since the methodology has changed compared to the analysis in Bendinger et al. (2024), we kept this section, and adapted it. Therefore, we prefer to keep the description of the ray tracing to ensure that it can be understood without referring to Bendinger et al. (2024).

**Figure 11 in Summary**

I am rather sceptical that this figure helps to provide a synthetic view of the results. I think the text is sufficient (and even clearer).

We decided to omit Figure 11 since overall it has led to more confusion than clarification.

**Others, general or specifics**

I do not think that internal tide needs to be hyphenated in the title.

According to the editorial service of Copernicus it does need to be hyphenated. This was also suggested for Part 1. We appreciate the reviewer's comment, though.

The introduction touches on various aspects of internal incoherence, but does not give much detail on what is known in the region. I understand that the regions have not received much attention until recently (in terms of IT dynamics and incoherence), but it could be seen as a synthesis of what is known. For example, I noticed that New Caledonia can be identified in the incoherent variance fraction estimates of Zaron et al. (2017) or Nelson et al. (2019).

Realistic numerical modeling: Nelson et al. 2019 suggests that >50 % of the SSH variance are explained by nonstationary semidiurnal internal tide. The fraction in Zaron 2017 is much reduced, but it was shown that the SSH variance estimate in wavenumber domain (as commonly done using conventional altimetry) is linked with limitations inherent in the sampling of altimeters which alias the tidal frequencies. The nonstationary fraction of the semidiurnal energy flux lies around 20 %, but becomes increasingly important with increasing distance to the generation sites (up to 80 %) before they diminish (Buijsman et al. 2017). We include part of this in the revised manuscript in Lines 47-48.

Eq. 1: Please briefly explain where this equation comes from (and/or add relevant references).

Former Equation 1 was motivated by Kang and Fringer (2012) and Lahaye et al. (2020). Eventually, we decided to show only the simplified baroclinic energy budget. However, we explicitly mention that the tendency of total energy (kinetic energy and available potential energy) is neglected while considering only hydrostatic pressure work contributions to the baroclinic energy flux. See Lines 146-150

L. 208–210: There is an incoherence here: depth-independent currents correspond only to a barotropic structure, as a mode-1 baroclinic current as a vanishing vertical integral (or average). My understanding is that using depth-independent currents for the ray-tracing approach is rather a simplification, because including the effect of the vertically sheared background current is complicated – unless one is willing to implement a much more complicated approach, as in Duda et al. 2018 or Guo et al (2023).

We agree with the reviewer. To avoid confusion we modified the sentence as follows: "The choice of depth-averaged currents is simplistic and relies on the assumption that vertically sheared background currents associated with mesoscale eddies do not alter the qualitative picture of ray trajectories that are obtained." See Lines 198-200

On l.304, the authors claim that local effects associated with local stratification changes are associated with a modulation of the amplitude of the baroclinic bottom pressure, while remote effects are associated with pressure phase variations. Is there any evidence or previous work to support this claim? While I suppose the former can be anticipated from linear theory (e.g. following St Laurent and Garrett 2002), I do not see a straightforward explanation for the impact of remote waves being associated only with phase variations.

We appreciate the reviewer's comment. Remote effects can induce phase and amplitude variations. Local effects induce amplitude variations only (with the phase remaining unchanged). This is explicitly mentioned in Lines 289-290. We also refer to Zilberman et al. (2011).

l.325: Again, how are variations in near-bottom stratification and bottom pressure related? Is this straightforward from the above equation for $p(z)$.

This is an important point to discuss. We included the following paragraph in the revised manuscript: "Pressure perturbations and stratification changes are directly linked assuming a second-order Taylor expansion of pressure around depth $z_0$:

$$p(z) \sim p(z_0) + \left.\frac{\partial p}{\partial z}\right|_{z_0} (z - z_0) + \left.\frac{\partial^2 p}{\partial z^2}\right|_{z_0} \frac{(z - z_0)^2}{2}, \tag{1}$$

In hydrostatic balance ($\frac{\partial p}{\partial z} = -\rho g$), and expressed via stratification ($\frac{\partial^2 p}{\partial z^2} = \rho_0 N^2$), this can be written as: $p(z) \sim p(z_0) - \rho g(z - z_0) + \rho_0 N^2 \frac{(z-z_0)^2}{2}$. By taking the time derivative and assuming adiabatic motion ($\frac{\partial \rho}{\partial t} = 0$), we obtain the relation:

$$\frac{\partial p}{\partial t} \sim \rho_0 \frac{\partial N^2}{\partial t} \frac{(z - z_0)^2}{2}. \tag{2}$$

In practice, this translates to decreasing (increasing) stratification, which corresponds to more widely (closely) spaced isopycnals leading to weaker (stronger) baroclinic pressure anomalies." See Lines 309-316

Figure 5 caption: missing parenthesis at end of first sentence.

This has been corrected.

l.409: the delay is in hours, not days. Also, the delays obtained are shorter in the south than in the north, which is somewhat contradictory with the fact that IT is more incoherent there.

The delay is indeed in hours. Note that with the new analysis we decided to omit the group arrival of semidiurnal rays to focus more on the cumulative refraction due to mesoscale stratification and/or currents.

**References (not mentioned in the manuscript**

- [**Dossmann et al. 2020**]: Dossmann, Y., Shakespeare, C., Stewart, K., & Hogg, A. McC. (2020). Asymmetric Internal Tide Generation in the Presence of a Steady Flow. *Journal of Geophysical Research: Oceans*, **125**, e2020JC016503.

- [**Dunphy & Lamb 2018**]: Lamb, K. G., & Dunphy, M. (2018). Internal wave generation by tidal flow over a two-dimensional ridge: energy flux asymmetries induced by a steady surface trapped current. *Journal of Fluid Mechanics*, **836**, 192–221.

- [**Nelson et al. 2019**]: Nelson, A. D., et al. (2019). Toward Realistic Nonstationarity of Semidiurnal Baroclinic Tides in a Hydrodynamic Model. *Journal of Geophysical Research: Oceans*, **124**, 6632–6642.

- [**St. Laurent & Garrett 2002**]: St. Laurent, L., & Garrett, C. (2002). The Role of Internal Tides in Mixing the Deep Ocean. *Journal of Physical Oceanography*, **32**, 2882–2899.

- [**Savage et al. 2020**]: Savage, A. C., Waterhouse, A. F., & Kelly, S. M. (2020). Internal tide nonstationarity and wave-mesoscale interactions in the Tasman Sea. *Journal of Physical Oceanography*, 1–52. `https://doi.org/10.1175/JPO-D-19-0283.1`

- [**Shakespeare 2020**]: Shakespeare, C. J. (2020). Interdependence of internal tide and lee wave generation at abyssal hills: global calculations. *Journal of Physical Oceanography*. `https://doi.org/10.1175/JPO-D-19-0179.1`

- [**Wunsch 2006**]: Wunsch, C. (2006). *Discrete Inverse and State Estimation Problems: With Geophysical Fluid Applications*. Cambridge University Press. `https://doi.org/10.1017/CBO9780511535949`

- [**Zaron & Egbert 2014**]: Zaron, E. D., & Egbert, G. D. (2014). Time-Variable Refraction of the Internal Tide at the Hawaiian Ridge. *Journal of Physical Oceanography*, **44**, 538–557.

---

## Author Comment (AC2)

Submitted to Ocean Science: Review of

**Regional modeling of internal-tide dynamics. Part 2: Tidal incoherence and implications for sea surface height observability**

**Reviewer #2**

The study describes the temporal variations of internal tide around New Caledonia from an energy perspective and serves as a companion to Bendinger et al. (2023). The authors conclude that mesoscale eddies contribute to the variations of internal tide generation, as shown through the decomposition of the incoherent term $C^{\mathrm{inc}}$ , and propagation, analyzed using a ray-tracing model. Specially, at the generation site (the near field), variations in conversion are linked to changes in baroclinic bottom pressure, which correlate positively with mesoscale-induced stratification changes. After propagation (the far field), regions with strong eddy activity exhibit enhanced incoherent tides and refraction in the propagation direction. The study also explores implications for SSH observability , highlighting that incoherent tides and the orientation of altimetry tracks can impact the separation of balanced and unbalanced motions.

Overall, the study is well-executed, providing solid evidence for the proposed mechanisms through quantified analysis. The discussion on the impact of altimetry track orientation on SSH wavenumber spectra is novel and relevant to the SWOT mission. However, the manuscript is somewhat lengthy, and certain sections could be condensed for clarity. Additionally, I have several questions and comments that would like to be addressed before publication.

First of all, we would like to thank the reviewer for taking the time to go through our manuscript, for the thorough comments and recommendations which helped improve clarity while pointing out the key findings of our study. We tried to shorten the manuscript as recommended, when possible. In the following, we will address the reviewer's comment and state where changes have been made in the manuscript.

**Introduction**

The recap for Part 1 (l.34-54) can be shorten by focusing on those related to incoherence.

The paragraph was slightly shortened. We prefer not to shorten it further since it is important for the rest of the paper, particularly for those readers which are not familiar with Part 1.

The energy dissipation is introduced in l.71-81, and understanding its temporal variation is stated as one of the objectives in l.106. However, the analysis of energy dissipation is only addressed at

the annual mean timescale and described in a single paragraph (l.241-251). The emphasis on the dissipation term in the introduction does not align with the following analysis focus.

Understanding temporal variations of dissipation was removed as one of the objectives. It is an objective in a broader sense, but as pointed out by the reviewer we did not intend to address this in our study. We insist on having it as a perspective (see Sect. 7.4). We included it also in the abstract: "Variations in conversion are not consistently proportional to those in energy flux divergence suggesting that variations in energy dissipation are linked to additional mechanisms that deserve further investigation." See Lines 12-14

**Section 2**

Equation 1 can be omitted to maintain relevance and save space.

Eventually, Equation 1 was omitted. However, we think that it is important to mention that Equation 1 in the revised manuscript neglects the tendency term of total (kinetic + potential) energy. Further, the energy flux considers only hydrostatic pressure work. See Lines 145-150

Equation 7, why is the energy flux intergraded from 0 instead of $\eta$.

We thank the reviewer for spotting this. It is indeed integrated from $\eta$. This is now corrected in Equation 3, 6, and 7 in the revised manuscript.

Equation 11, the right parathesis and "dt" for $D_{bc}^{coh}$ term are missing.

We thank the reviewer for spotting this. This is now corrected in Equation 10 in the revised manuscript.

At the end of Section 2.2, the authors state that the $D_{bc}^{inc}$ represents the overestimated portion of $D_{bc}^{coh}$. A similar statement appears in Section 3 (l. 246-247). However, I think the $D_{bc}^{inc}$ includes both the overestimated coherent portion AND the actual dissipation from the incoherent tide. The presence of real incoherent dissipation can be verified by the net incoherent dissipation at North (1). Nevertheless, the conclusion regarding the overestimation remains unchanged. Section 2.3 is similar to Bendinger et al (2024) and can be shorten.

The reviewer is absolutely right. We forgot to mention this. We modified the associated paragraph in Section 2.2: "While $D_{bc}^{D2}$ accounts for actual energy dissipation, $D_{bc}^{inc}$ consists of both the fraction by which $D_{bc}^{coh}$ is mistakenly associated with true energy dissipation (or the error by which energy dissipation in $D_{bc}^{coh}$ is overestimated) and incoherent energy dissipation." (see Lines 190-193). Similarly, we modified this paragraph in Section 3: "$D_{bc}^{inc}$ consists of incoherent energy dissipation and energy transferred from the coherent tide to the incoherent tide." (see Lines 231-232).

We prefer to keep Section 2.3 to ensure that the ray methodology can be understood without referring to Bendinger et al. (2024). Furthermore, by addressing a comment from reviewer #1, the methodology has changed. Briefly, Section 5 has slightly been enriched by considering effects of stratification as well to quantify the extent by which the rays are refracted by mesoscale currents

and stratification. Section 2.3 and 5 were accordingly adapted.

**Section 4**

It is unusual to have only one subsection. I suggest reorganizing the structure. The same applies to Section 5.

The structure has been reorganized: Section 4 is now splitted in two subsections. The subsection for Section 5 is omitted.
4 What drives semidiurnal barotropic-to-baroclinic energy conversion variability?
4.1 Coherent vs incoherent contributions
4.2 Mesoscale-eddy-induced conversion variations
5 Mesoscale-eddy-induced refraction of tidal beams lead to increasing tidal incoherence in the far field

The explained variability is expressed as (%) in text (e.g. l.259-260) while as decimal (0-1) in Table 2. Please ensure consistency.

The explained variability is now also expressed as decimal in the text when assigned to $\gamma$.

L.315: "The negative conversion/bottom pressure amplitude anomalies in Fig. 5b". Should likely refer to "Fig. 5a"?

Yes, well spotted. This paragraph was slightly adapted for the sake of clarity to the following: "The monthly time series of bottom stratification $N^2(-H)$ (extracted from the bottom most grid cell), mesoscale SLA, and mesoscale EKE (similarly computed to Sect. 3.2 in Part 1) suggest that conversion variations through $dP_A$ are linked with mesoscale-eddy-induced stratification changes (Fig. 5a and b)." See Lines 301-303

L.327: "In phase with the local tidal forcing, $p_{\mathrm{bc}}^{\mathrm{inc}}(-H)$ induced..." If $p_{\mathrm{bc}}^{\mathrm{inc}}(-H)$ is "in phase", why it is incoherent?

Local effects are not expressed by phase variations as stated in Lines 289-290. In other words, local effects are linked with amplitude variations only with the phase remaining unchanged. We refer to Zilberman et al. (2011). In our case, amplitude variations are linked with mesoscale-eddy-induced stratification changes.

**Figure 3**

The bathymetric labels are too small to be clearly visible.

Bathymetry labels have been increased in size in Figure 3 and 4.

**Figure 5**

5(a): What is the physical meaning of the negative values for the ratios of $C^{\text{cross1}}/C^{\text{D2}}$ on the y axis?

Please note that these ratios represent anomalies referenced to the coherent conversion, which explains why there are positive and negative values. Positive (negative) values for this ratio correspond to an increase (decrease) of conversion relative to the coherent conversion.

**Table 3**

The unit for "delay" should be [days].

In fact, the delay is indeed in [hours]. See the comment below for more details.

**Section 5**

The "group arrival time" is not clearly defined in Section 5.1. Based on the caption in Table 3, which states "equivalent to 500 km", I assume the "group arrival time" refers to the time taken to propagating 500 km. If so, for South (2) domain, both mode-1 and mode-2 tide propagate faster with the mesoscale currents, which is consistent with the negative "delay" time. If my understanding is correct, the statement in l.409-410, "Mode 2 is substantially more delayed than mode 1", is incorrect for the South (2) domain, as both mode-1 and mode-2 arrive earlier, with negative delayed time.

This statement of mode-2 being more delayed is indeed wrong. Negative delay would imply that mode-2 is faster compared to mode-2 without currents.

As stated above, Section 5 has been modified to meet suggestions from reviewer #1. With the new analysis, we decided to omit the group arrival of semidiurnal rays while focusing on their cumulative refraction due to mesoscale stratification and/or currents. The latter was achieved by integrating the orientation change or angle of the group velocity vector along the propagation path. Note that we integrate absolute values of angular deviation. The group arrival delay relative to the reference ray with annually averaged stratification and currents is relatively small: a maximum of 1-2 hours during a propagation of >2 days. However, we plotted in Figure 8 the daily location (day 1-2) of the rays. Also, note that we omitted the discussion on mode-2 since the semidiurnal energy flux is strongly dominated by mode 1.

I also speculate on the "delay" for South (2) domain. The standard deviation is large enough to cause the "delay" time to switch signs.

In fact, the sign of delay depends whether the ray is refracted northward or southward relative to the reference ray. Since the refraction is of stochastic nature, the mean delay (averaged over all rays) can be close to zero with a standard deviation, which is large enough to cause the delay to switch signs as pointed out by the reviewer.

**Section 6**

The paragraphs before 6.1 (l.421-445) serve as a recap in Part 1 and an introduction, so they can be condensed. The same applies to l.459-465 in Section 6.2.

l.421-445 were condensed (see Lines 405-419 in the revised manuscript). Particularly, the results recap from Part 1 was omitted since it is already mentioned in Section 2.1 (see Lines 133-139). l.459-465 are already very concise, but we decided to generalize the findings from Part 1.

Below are some minor comments, which I leave to the authors' discretion to consider.

**Vague pronoun reference:** please explicitly state the references for clarity.
L.10: "it"
it" is now replaced with "incoherent conversion"

L.77: "this can have..."
"this" is now replaced with "energy dissipation associated with the incoherent tide"

L.141: "their Fig. 13 a-d"
"their Fig. 13a-d" is now replaced with "see Fig. 13a-d in Part 1"

L.259: "it explains"
"it" is now replaced with "the coherent conversion"

**Overuse of "i.e.":** here are some suggested replacements.
L.141: "... Caledonia, including the location..."
This suggestion was taken into account.

L.163: "... incoherent parts for u and the pressure perturbation p"
This suggestion was taken into account.

L.255: "... spring-neap cycle, driven by the interaction of M2..."
This suggestion was taken into account.

**Formatting and grammar corrections** We thank the reviewer for comments on formatting and grammar. They were all taken into account, unless specified otherwise.

L.5: "...from coherence, in...", add space after the comma.
This suggestion was taken into account.
L.8 and l.255: The phrase "astronomically forced fortnightly modulated spring- neap cycle" is wordy and grammatically incorrect, which reduces readability. Consider a revision.
"astronomically forced fortnightly modulated spring-neap cycle" is now replaced with "interaction of M2 and S2 barotropic tidal currents" and "...due to the spring-neap cycle, driven by the interaction of M2 and S2 tidal constituents" See Lines 6-7 and Lines 240-242, respectively
L.9: use an "em dash" rather than a hyphen. The same applies to others in the manuscript.
Thank you for this comment. We were not aware of the usage of the "em dash". All hyphens were replaced by the "em dash".
L.29: "semi-analytical theory"

This suggestion was taken into account.
L.45: "representative to the coherent"
We believe that "representative of" is the correct usage here, but we will check with the Copernicus editorial service during proofreading.
L.57: I think "near-field" and "far-field" that have a hyphen in between, are commonly used as adjectives, rather than nouns.
We removed all hyphens in "near-field" and "far-field", unless it is used as an adjective.
L.64-65: suggested revision "The mechanisms governing the temporal variability of internal tide vary geographically, and cannot be generalized as the importance ..."
This suggestion was taken into account.
L.67-69: suggested revision "New Caledonia is a particularly challenging region as it is a hot spot of internal tide generation and a region of strong mesoscale variability, making it potentially ..."
This suggestion was taken into account.
L.94: "estimate the length scale at which unbalanced motions ..."
This suggestion was taken into account.
L.186: "taken from the harmonic analysis and vertical mode ..."
This suggestion was taken into account.
L. 215: missing the right parathesis after "(see Fig.1)"
This suggestion was taken into account.
L. 220: "mimic"
Section 2.3 was rewritten, and the associated sentence was removed.
L.239: "distance from the generation"
This suggestion was taken into account.
L.247: "This accounts for 10%, 9% ..."
This suggestion was taken into account.
L.276 (Figure 3 caption): The second sentence lacks a verb.
The second sentence was changed to "Explained variability shown for..."
L.280: "Similarly to the analysis above, we show in the South (2) domain, the contribution of different terms that make up $C^{\mathrm{inc}}$ ..."
This suggestion was taken into account.
L.282-283: "While the three terms feature similar amplitudes, their spatial patterns differ." This suggestion was taken into account.
L.283: "Based on the area-integrated explained ... "
This suggestion was taken into account.
L.317 and 319: "compute 5-day mean" and "period of 180 days"
Note that the methodology description is now in the caption of Figure 5. We will double check with the Copernicus editorial service during proofreading. For Part 1, it was suggested to use the notation of "5-d mean" and "180 d".
L.324: "conversion and mesoscale variability ...", which conversion?
The term conversion refers to barotropic-to-baroclinic energy conversion throughout the manuscript. Note that the associated sentence was slightly rewritten/rephrased. See Lines 309-321.
L.335: "pressure amplitude variations are very pronounced, suggesting the influence of the local effects"
This suggestion was taken into account.
L.379: "closely correlated to that of semidiurnal"
This suggestion was taken into account.
L.612: "concerns the impact of conversion variability on outward energy propagation and local energy dissipation"? The logic between these three terms in Equation 2 is unclear to me.

We slightly rewrote the paragraph to the following to improve clarity: "One open question arising from our analysis is how variability in barotropic-to-baroclinic energy conversion influences both the outward propagation of internal-tide energy and local dissipation. According to the baroclinic energy budget (Equation 1), dissipation is defined as the residual between conversion and energy flux divergence. We therefore ask: do variations in conversion directly translate into proportional changes in energy flux divergence—and, by extension, in dissipation—or are they partially decoupled? For instance, does increased conversion always imply stronger outward flux and higher dissipation (Falahat et al. (2014)?". See Lines 585-590. This serves as a perspective. According to Figure 12 in the revised manuscript, variations in conversion and energy flux divergence can be decoupled.